# Mercury's anomalous magnetic field caused by a symmetry-breaking self-regulating dynamo

Futoshi Takahashi[1], Hisayoshi Shimizu[2] & Hideo Tsunakawa[3]

The discovery of Mercury's unusually axisymmetric, anomalously axially offset dipolar magnetic field reveals a new regime of planetary magnetic fields. The cause of the offset dipole remains to be resolved, although some exotic models have been proposed. Deciphering why Mercury has such an anomalous field is crucial not only for understanding the internal dynamics, evolutionary history and origin of the planet, but also for establishing the general dynamo theory. Here we present numerical dynamo models, where core convection is driven as thermo-compositional, double-diffusive convection surrounded by a thermally stably stratified layer. We show that the present models produce magnetic fields similar in morphology and strength to that of Mercury. The dynamo-generated fields act on the flow to force interaction between equatorially symmetric and antisymmetric components that results in north-south asymmetric helicity. This symmetry-breaking magnetic feedback causes the flow to generate and maintain Mercury's axially offset dipolar field.

[1] Department of Earth and Planetary Sciences, Faculty of Science, Kyushu University, 744 Motooka, Nishi-ku, Fukuoka 819-0395, Japan. [2] Earthquake Research Institute, University of Tokyo, 1-1-1 Yayoi, Bunkyo-ku, Tokyo 113-0032, Japan. [3] Department of Earth and Planetary Sciences, Tokyo Institute of Technology, 2-12-1 Ookayama, Meguro-ku, Tokyo 152-8551, Japan. Correspondence and requests for materials should be addressed to F.T. (email: takahashi.futoshi.386@m.kyushu-u.ac.jp)

I n-orbit observations by the MErcury Surface, Space ENvironment, GEochemistry and Ranging (MESSENGER) spacecraft confirms that Mercury currently possesses a global magnetic field generated by convective motions in the liquid iron core through dynamo processes[1–3]. Mercury's field intensity is about 1% of the Earth's field intensity at the surface (~700 nT), and has shown very weak secular variation over the past 40 years[4]. Moreover, MESSENGER has revealed the unusual morphology of Mercury's magnetic field, which is unlike that of any other planetary magnetic field: strongly axisymmetric dipolar fields with a dipole tilt angle <0.8° and the magnetic equator displaced northward by 0.2 $R_H$, where $R_H$ = 2440 km is the Hermean radius[1–3]. Note that the MESSENGER measurements of the magnetic field were strongly biased towards the northern hemisphere, and that a smaller displacement of 0.14 $R_H$ was reported in an analysis with a larger dataset[5].

Based on the co-density approach of treating thermal and compositional convection together[6], the weak field and its low secular variation can be explained by dynamo models incorporating a thermally stably stratified layer beneath the core mantle boundary (CMB), through which small-scale, high-frequency components deep inside the core are attenuated due to the skin effect[7,8]. However, previous models have hitherto had difficulty reproducing the unusual morphology without rather speculative CMB boundary conditions[9,10].

There are two key issues limiting these earlier findings: the co-density formulation, instead of the treatment of the flow as double-diffusive convection, was used in the anticipation of turbulent diffusivity, and this formulation may be invalid in a thick stably stratified layer[11]; and the unique core crystallization due to the pressure–temperature condition of small bodies such as Mercury was not taken into account, but could result in compositional convection in a non-Earth-like manner, depending on the unknown core sulfur concentration[12,13].

Three representative mechanisms have been proposed as potential drivers of the compositional convection in Mercury's core: a bottom-up (BU), top-down (TD), and snow-layer (SL) mode[14]. The BU mode is powered by either ejection of an Earth-like light element from the inner core boundary (ICB) or a Ganymede-like floatation of light FeS solid from an Fe–FeS alloy on the FeS-rich side of the eutectic[12]. The TD mode corresponds to precipitation of an Fe-rich solid as iron snow from the CMB or the bottom of the stably stratified layer[15]. The SL mode represents the case where iron snow occurs at a certain depth of the core[16]. The combined effect on Mercury's dynamo of double-diffusive convection and a core crystallization regime has not yet been explored.

Here, we present numerical dynamo models driven by thermo-compositional, double-diffusive convection in a rotating spherical shell that reproduce all the characteristic features of Mercury's magnetic field (see Methods). As in our previous study[17], the diffusivity contrast between thermal and compositional diffusivities is 10, which is rather small compared with those expected in planetary cores. Even such a small difference could yield the magnetic fields different from those in the co-density model[11,17]. In the present model, a thermally stably stratified layer is imposed in the upper half of the core (Fig. 1). The heat/compositional flux is assumed to be fixed on the ICB and CMB. In particular, the zero-heat-flux condition is applied to the ICB to minimize the effects of bottom heating or maximize those of internal heating to drive thermal convection[9]. In all, we performed 12 runs for the BU, TD, and SL modes, where thermal and compositional driving forces (Rayleigh numbers) were varied, while other parameters were fixed. The parameters used in the present study are described in detail in the Supplementary Table 1.

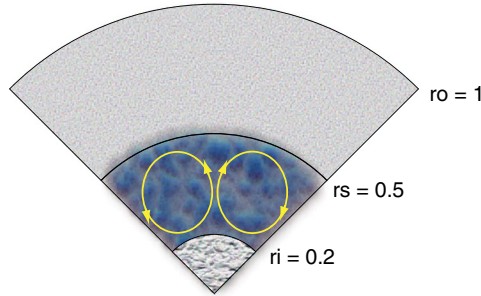

ro = 1

rs = 0.5

ri = 0.2

**Fig. 1** Model geometry in units of core radius. The radius of the solid inner core is $r_i$ = 0.2. The interface of the thermally stably stratified layer is set at $r_s$ = 0.5. Above $r_s$, the thermally stably stratified region extends to the core mantle boundary, $r_o$ = 1

## Results

**Comparison with observations**. The radial component of the model magnetic field at the planetary surface is dominantly axisymmetric, dipolar, and asymmetric about the equator, which is consistent with the observations as well as with the strength of the field (Fig. 2a, b). The average dipole offsets from the center are 0.14 $R_H$ and 0.2 $R_H$ in the cases of BU1 and BU2, respectively (Supplementary Table 1); these values are comparable with the observed ones[3,5]. In order to explain the dipole offset, the Hermean-centered quadrupole component, amounting to 40% of the Hermean-centered axial dipole, is required in the case of BU2 and Mercury's field[3], whereas the quadrupole component is 28% of the dipole in BU1[5]. On the other hand, the TD and SL models result in field morphologies dissimilar to the observed ones— namely, an unacceptably strong and equatorially antisymmetric morphology (Fig. 2c) and an overly weak and dominantly non-dipolar morphology (Fig. 2d). These features are also confirmed by the magnetic power spectrum in terms of spherical harmonic degree and order (Fig. 2e, f). Moreover, the mean dipole tilt angle from the spin axis is less than 1°, and slow magnetic field variation like that of Mercury[2,4] is produced in BU1 (Supplementary Fig. 7 and Supplementary Table 1).

**Formation of the offset dipolar field**. The dipole offset results from the biased dynamo process in the core of the northern hemisphere (Fig. 3a), which is prompted by the equatorially asymmetric flow structure (Fig. 3b–d). The asymmetric structures in the convection vortices and magnetic field are temporally stable. The hemispherically averaged relative axial helicity (RAH) of convection is an important quantity with respect to the resultant magnetic field morphology[18]. In BU1, the time-averaged |RAH| is 0.42 for the northern hemisphere and 0.34 for the southern hemisphere (Supplementary Table 1), and these values differ significantly. It has been suggested that the north–south asymmetry in RAH is caused by the antisymmetric mode of convection, the kinetic energy of which is merely ~10% of the total[9].

Analysis of the axial helicity partition indicates that the symmetric and antisymmetric flows interact to yield the hemispherical bias in RAH (Supplementary Fig. 2a, b). Other dynamo models driven by TD and SL but failing to have dipole offset show an insignificant difference of RAH between hemispheres despite having a fraction of the antisymmetric flow components similar to the BU model, indicating a negligible interaction between different flow modes (Supplementary Fig. 2c–f). More importantly, we found in the present study that an interactive flow structure that yields dipole offset is maintained by the dynamo-generated magnetic field itself. By switching off

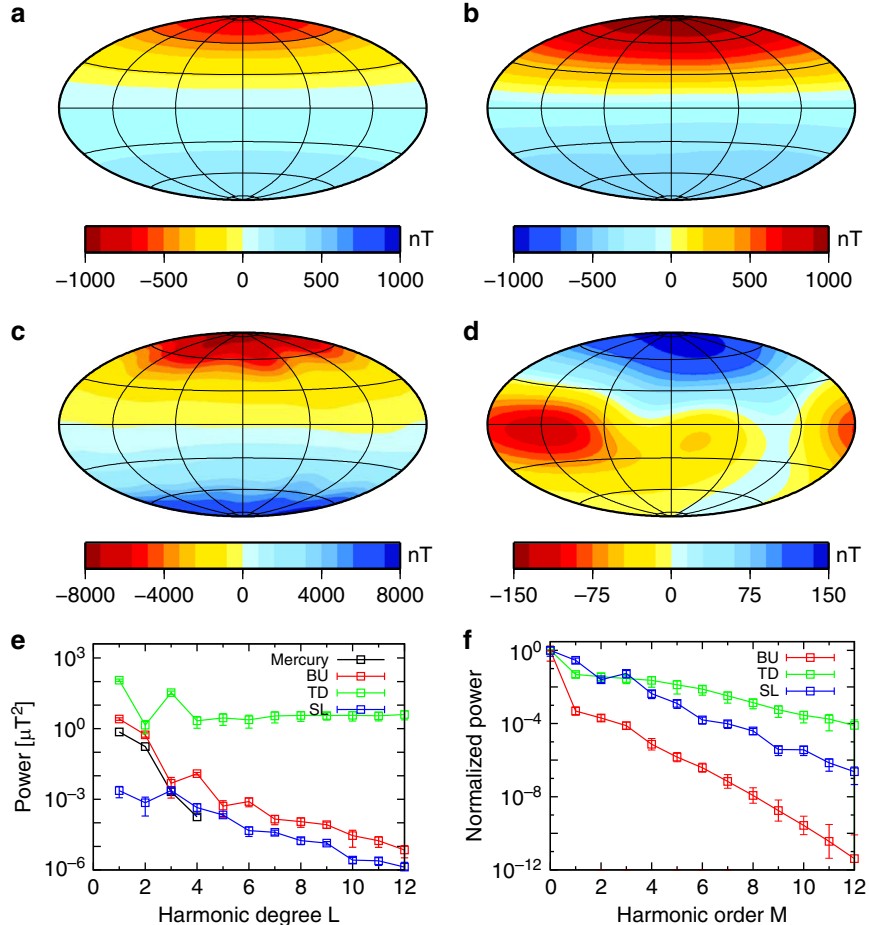

**Fig. 2** Snapshots of the radial magnetic field distribution and time-averaged power spectrum with standard deviation at Mercury's surface. **a** MESSENGER observation[2]; **b** BU1; **c** TD1; **d** SL1. The magnetic field is scaled by $(2\rho\mu\eta\Omega)^{1/2} = 0.14$ mT, where $\rho = 6980$ kg m$^{-3}$ is density[29], $\mu = 4\pi \times 10^{-7}$ H m$^{-1}$ is magnetic permeability in a vacuum, $\eta \sim 1$ m$^2$ s$^{-1}$ is magnetic diffusivity[43], and $\Omega = 1.2 \times 10^{-6}$ s$^{-1}$ is planetary rotation rate. The thickness of the mantle is assumed to be 590 km[7, 8, 11]. Color scale is reversed in **b** for the purpose of illustration. The magnetic equator corresponds to where the radial component is zero. **e** Power spectrum vs. spherical harmonic degree $L$ in units of $\mu$T$^2$ in cases of observation (black), BU1 (red), TD1 (green), and SL1 (blue). Error bars represent one standard deviation. The close agreement between BU1 and observation up to $L = 3$ (octupole) component is remarkable. **f** Power vs. spherical harmonic order $M$ normalized by the $M = 0$ (axisymmetric) component. The axisymmetric component in BU1 dominates the other components by a factor of at least 1000

the feedback effect of the magnetic field on convection in BU1K (Helicity analysis in Methods), a nearly perfectly north–south symmetric flow structure ensues (Fig. 4a), and the quadrupolar magnetic field is generated as a result of kinematic dynamo action (Fig. 4b). The antisymmetric flow component rapidly declines with time (Supplementary Fig. 8), indicating that the interacting antisymmetric flow mode is driven by the magnetic field. Thus, we conclude that a Mercury-like offset dipolar field is generated by a dynamo, with symmetry-breaking magnetic feedback to yield helicity bias, which we call self-regulation (SR).

## Discussion

The present models show that, without taking any speculative external forcing into account, a Mercury-like magnetic field could result from a spontaneous dynamo mechanism driven by thermo-compositional convection, in which the compositional buoyancy is caused by the BU compositional process and thermal buoyancy mostly by the internal heating in the liquid core. In a previous study using double-diffusive convection, thermal convection is driven from the bottom (i.e., non-zero ICB heat flux due to latent heat release; Supplementary Fig. 4) with a larger inner core,

resulting in stronger, multipolar dynamos[11]. According to runs of BU1–4 and those in a previous study[11], Mercury-like magnetic morphology is found in cases of zero ICB heat flux with a small inner core and modest flow vigor in terms of a magnetic Reynolds number of ~100 (Supplementary Table 1), which is close to the estimate for Mercury's core[9].

Each model has a velocity field of clearly different structure (Supplementary Fig. 5): convection confined within a convectively unstable region in the case of BU, fingering-type convection in a thermally stably stratified layer in TD[11], and faintly layered convection in SL[16]. The radial distributions of the velocity and magnetic fields show the prominence of the axisymmetric toroidal field in BU1 around the stratification boundary (Supplementary Fig. 6). The magnetic field within the convective region is strong enough for SR to work, particularly near the stable–unstable stratification boundary, whereas the weak magnetic field at the surface is due to attenuation associated with the skin effects of the stably stratified layer.

Therefore, in terms of the mechanisms involved in the generation and maintenance of Mercury-like magnetic field morphology, the key findings of the present work are as follows: the axial helicity distribution biased to one hemisphere is due to

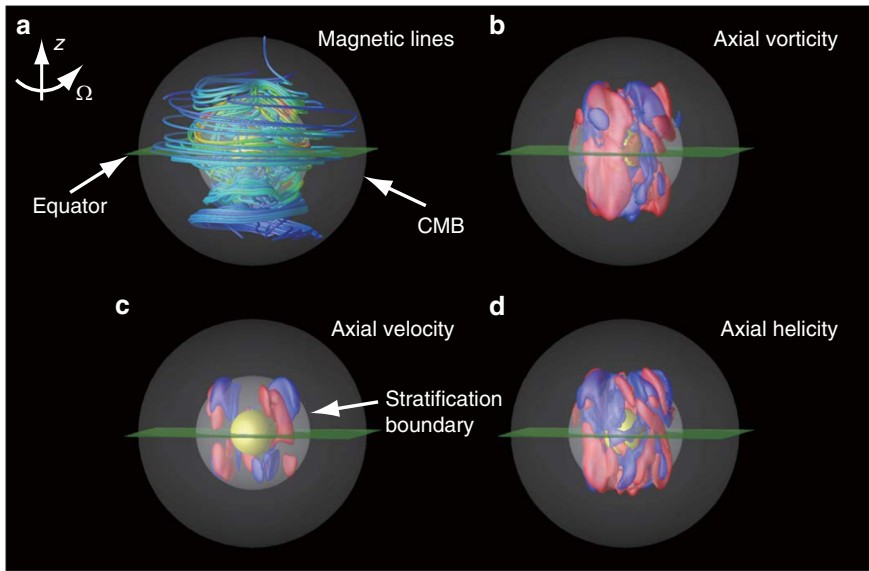

**Fig. 3** Snapshot of the internal structures of the magnetic field and flow. The snapshot is taken at the same time as in Fig. 2b in the near-equatorial view. The equatorial plane is in green, while the solid inner core is represented by a yellow sphere. The direction of rotation is aligned with the z-axis. Spherical darker regions represent the thermally stably stratified layer. **a** Magnetic field structure represented by magnetic field lines. Red (blue) lines represent the strong (weak) magnetic field. **b** The axial vorticity represented by isosurfaces of ±0.03 yr$^{-1}$. Red denotes anti-cyclone, and blue denotes cyclone. **c** The axial flow represented by isosurfaces of ±3 km yr$^{-1}$. Red (blue) denotes northward (southward) flow. **d** The axial helicity distribution represented by isosurfaces of ±2300 km yr$^{-2}$. Red (blue) denotes a positive (negative) value

non-linear interaction of the equatorially symmetric and anti-symmetric flow modes; and such interaction is prompted by the Lorentz force powering the antisymmetric flow mode as an SR process. These findings are in contrast to a previous study which explained the helicity asymmetry in terms of linear superposition of the symmetric and antisymmetric modes[9]. The asymmetric flow components also alter convective heat/compositional flux profiles in an asymmetric manner biased to the northern hemisphere (Supplementary Fig. 9a). It should be mentioned that in spite of the antisymmetric flow mode being much smaller than the symmetric one here, their symmetry-breaking interaction could play a decisive role in generating an asymmetric dynamo[19]. Taking into account the fact that the action of the Lorentz force enhancing the helicity tends to prefer dipolar fields over quadrupolar fields[20,21], and that the quadrupolar dynamo is obtained in the kinematic run (Fig. 4b), our results suggest that a convection that prefers the quadrupolar field in the absence of SR may require the hemispherical magnetic morphology. If so, feedback of the large-scale strong azimuthal toroidal field may play a role in generating the dipole component[22,23].

Therefore, we next examined this possibility. The axisymmetric components of the magnetic and velocity fields in Fig. 5 show distinctive differences in structure. In the case of BU1, the asymmetric meridional circulation and zonal flow indicate an SR of the magnetic field acting in the northern hemisphere (Fig. 5a), whereas the almost symmetric flow structures in TD1 and SL1 denote a negligible SR effect on convection (Fig. 5b, c). Compared to BU1K (Fig. 5d), it is clear that the SR alters the flow and magnetic field structures in BU1. Note also that the azimuthal (toroidal) magnetic field generation around the edge of the convection columns and stratification boundary at mid-latitudes in the northern hemisphere is well correlated with the generation of the axial dipole and axial quadrupole components (Supplementary Fig. 10a). In addition, the zonal flow structure is deformed in the northern hemisphere by the SR. Without SR, the symmetric flow structure would appear as in BU1K, and then the symmetric toroidal field would be regenerated only near the

equator (Supplementary Fig. 10d). Eventually, the axial dipole component would also disappear. Although the field generation in the northern polar region is also remarkable, SR effects are unclear there. On the other hand, in TD and SL, magnetic field generation occurs in an almost symmetric way, and therefore there seems to be no SR in operation to feed the magnetically driven antisymmetric flows and break equatorial symmetry. Thus, it is suggested that generation of a biased axisymmetric toroidal field around the stably stratified layer is an important factor if the SR is to work.

To test whether double-diffusive treatment instead of co-density is necessary for the present result, we compared the BU1 to the case of BU1C (Supplementary Note 1) corresponding to the co-density approach, where thermal and compositional diffusivities were set to be equal. The BU1C results in a dipolar magnetic field without significant dipole offset, whose strength is much larger than that of Mercury (Supplementary Fig. 11 and Supplementary Table 1). As shown in a previous study[11], dynamos by double-diffusive convection and co-density could be different. Here, it is also demonstrated that double diffusion is an important factor for a Mercury-like field, although the complex physical processes involved must be elucidated in future research.

Asymmetric hemispherical dynamos have also been examined with thermal convection in other bodies such as the Sun[24], Mars[25–27], and Ganymede[15]. In the case of Sun-like models, the magnetic field contains substantial non-axisymmetric components, and shows wavelike periodic reversals. For the Mars-like models, a north–south asymmetric, heterogeneous outer boundary condition is imposed to create an asymmetric convection structure and magnetic field. In most of these models, the complicated magnetic field morphology contains non-axisymmetric components. Under homogeneous outer thermal boundary conditions, the asymmetric dynamo can be generated by the equatorially antisymmetric, axisymmetric (EAA) flow[28]. Because the EAA mode occupying the kinetic energy comparable with that in the symmetric mode occurs spontaneously as a result of thermal instability, the SR effect is not required to form the asymmetric

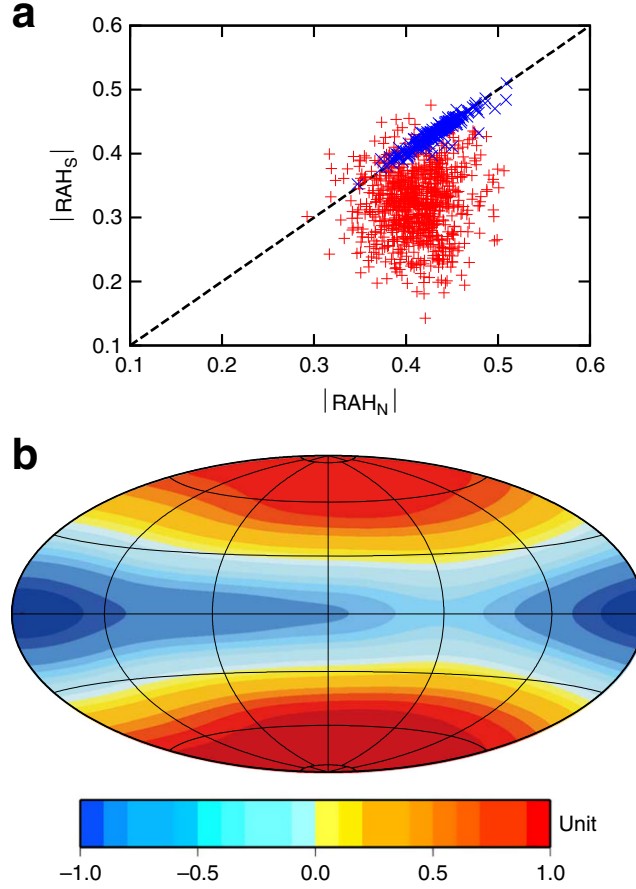

**Fig. 4** Effects of symmetry-breaking feedback of the magnetic field on convection to yield hemispherical bias in RAH and the offset dipolar field. **a** Plot of the absolute values of RAH in the northern hemisphere, $|RAH_N|$, vs. those in the southern hemisphere, $|RAH_S|$, in BU1 (red) and the corresponding kinematic run BU1K (blue), where the Lorentz force is removed from the momentum equation. Symbols represent those values at different times. When convection is perfectly symmetric with respect to the equator, $|RAH_N|$ and $|RAH_S|$ lie on the dashed line with a slope of unity as in the kinematic run. **b** Snapshot of the normalized radial magnetic field distribution at the planetary surface in the run BU1K. Unit is normalized by the maximum value, since the case is supercritical

structure. Dynamo models focusing on Ganymede by TD in the co-density formulation, where no SR seems to manifest, show a regime diagram between the input parameters, stable layer thickness, and magnetic field morphology[15]. The models mentioned should be investigated to see what the most probable mechanism is to spontaneously form a Mercury-like magnetic field. In this regard, whether or not the present findings could be extended towards the parameter regime appropriate to the planetary core remains to be examined. These are issues to be studied by a broader range of parameter survey.

The buoyancy source distribution in our Mercury-like dynamos has implications for the chemical composition and evolution of Mercury's core. Since thermal convection due to internal heating is preferred to that due to bottom heating from the latent heat release upon inner core solidification, slow cooling of the core, retarded inner core growth, and resultantly small inner core are suggested[29,30]. A considerable amount of radioactive heat source, such as potassium and uranium, would then be required to drive thermal convection, maintain a stably stratified layer, and keep the inner core small[31,32]. In this circumstance, compositional convection driven by the BU process due to inner core

growth could be modest, consistent with the moderate values of the magnetic Reynolds number in our models. However, in order to form a thermally stably stratified upper layer and a deep convectively unstable layer, there are some issues to be considered regarding the mutual compatibility of buoyancy sources in the BU models. In the case of homogeneously distributed internal heat sources, the heat flux varies in proportion to the radius. Then there is no crossover between the adiabatic and super-adiabatic heat flux, and the entire fluid core is either stable or unstable. In addition, the zero-heat flux condition at the ICB is not compatible with compositional flux. To justify the model assumptions, it could be plausibly argued that the ICB heat flux due to latent heat release is very close to the adiabatic one, or that the radioactive elements are concentrated only in the deep layer, although no obvious justifications are available for the model assumptions. Another possibility could be that the adiabatic temperature gradient varies super-linearly with the radius because of the strong pressure dependence of the thermal expansion coefficient, so that the adiabatic heat flux can exceed the actual heat flux in the upper parts of the core, but falls short of it in the deep parts. These arguments would need verification based on the thermodynamic properties of iron and structural models for Mercury[14,33].

A thermal and magnetic evolution model with an Fe-Si core allowing formation of a thermally stably stratified layer yields a present-day inner core larger than 800 km, and strong multipolar magnetic fields[33]. These findings may suggest that the core contains some amount of silicon, as well as a low concentration of sulfur (<6%), which prevents TD/SL convection by iron snow[14,34,35]. If so, these facts would pose additional constraints on the thermal and magnetic evolution of Mercury[36].

The present models could be tested by good-coverage observation in the next BepiColombo mission[37]. This would allow us to conduct a detailed comparison between the improved data of the internal magnetic field and the predictions derived from numerical dynamo models of Mercury.

## Methods

**Numerical modeling of a planetary dynamo.** We consider the convective motion of an electrically conducting, incompressible Boussinesq fluid in a rotating spherical shell of inner and outer radii $r_i$ and $r_o$. In most cases, the aspect ratio is $\chi = r_i/r_o = 0.2$, while an Earth-like value is $\chi = 0.35$. The spherical shell is rotating about the z-axis at an angular rotation rate $\Omega$. The governing equations described in non-dimensional form are the Navier–Stokes equation for the velocity field $\mathbf{u}$ and non-hydrostatic pressure $P$, induction equation for the magnetic field $\mathbf{B}$, heat transport equation for the temperature $T$, and transport equation for the light element concentration $C$. Length is scaled with the shell thickness $D = r_o - r_i$. Time is scaled with the viscous diffusion time $D^2/\nu$, where $\nu$ is the viscosity. The velocity is scaled with $\nu/D$. The magnetic field is scaled with $(2\rho\mu\eta\Omega)^{1/2}$, where $\rho$ is the density, $\mu$ is the magnetic permeability in vacuum, and $\eta$ is the magnetic diffusivity. The temperature and concentration of light elements are scaled with $h_o^T D$ and $h_i^C D$, where $h_o^T$ is the reference CMB temperature gradient without stable stratification and $h_i^C$ is the reference ICB compositional gradient. In some models, temperature and light element concentration are scaled using the reference ICB temperature gradient and CMB compositional gradient.

Non-dimensional parameters of the dynamo models are the thermal Prandtl number ($Pr^T = \nu/\kappa_T = 0.1$, where $\kappa_T$ is the thermal diffusivity), the compositional Prandtl number ($Pr^C = \nu/\kappa_C = 1$, where $\kappa_C$ is the compositional diffusivity), the magnetic Prandtl number ($Pm = \nu/\eta = 3$), the Ekman number ($E = \nu/2\Omega D^2 = 10^{-4}$), the thermal Rayleigh number ($Ra^T = \alpha g h_o^T D/2\Omega\nu$, where $\alpha$ is the rate of thermal expansion, and $g$ is the gravitational acceleration at CMB), and the compositional Rayleigh number ($Ra^C = \beta g h_i^C D/2\Omega\nu$, where $\beta$ is the rate of compositional expansion).

The non-dimensional reference temperature profile without a stably stratified layer $dT_o/dr$ is described as

$$\frac{dT_o}{dr} = -\frac{\varepsilon_T}{3}r + \frac{a_T}{r^2}, \tag{1}$$

where $\varepsilon_T = 3r_o^2/(r_o^3 - r_i^3)$ represents a uniformly distributed volumetric internal heat source, and $a_T = r_i^3 r_o^2/(r_o^3 - r_i^3)$. In the present case of $\chi = 0.2$, we have $\varepsilon_T = 2.42$ and $a_T = 1.26 \times 10^{-2}$. Based on the profile, a thermally stably stratified layer is

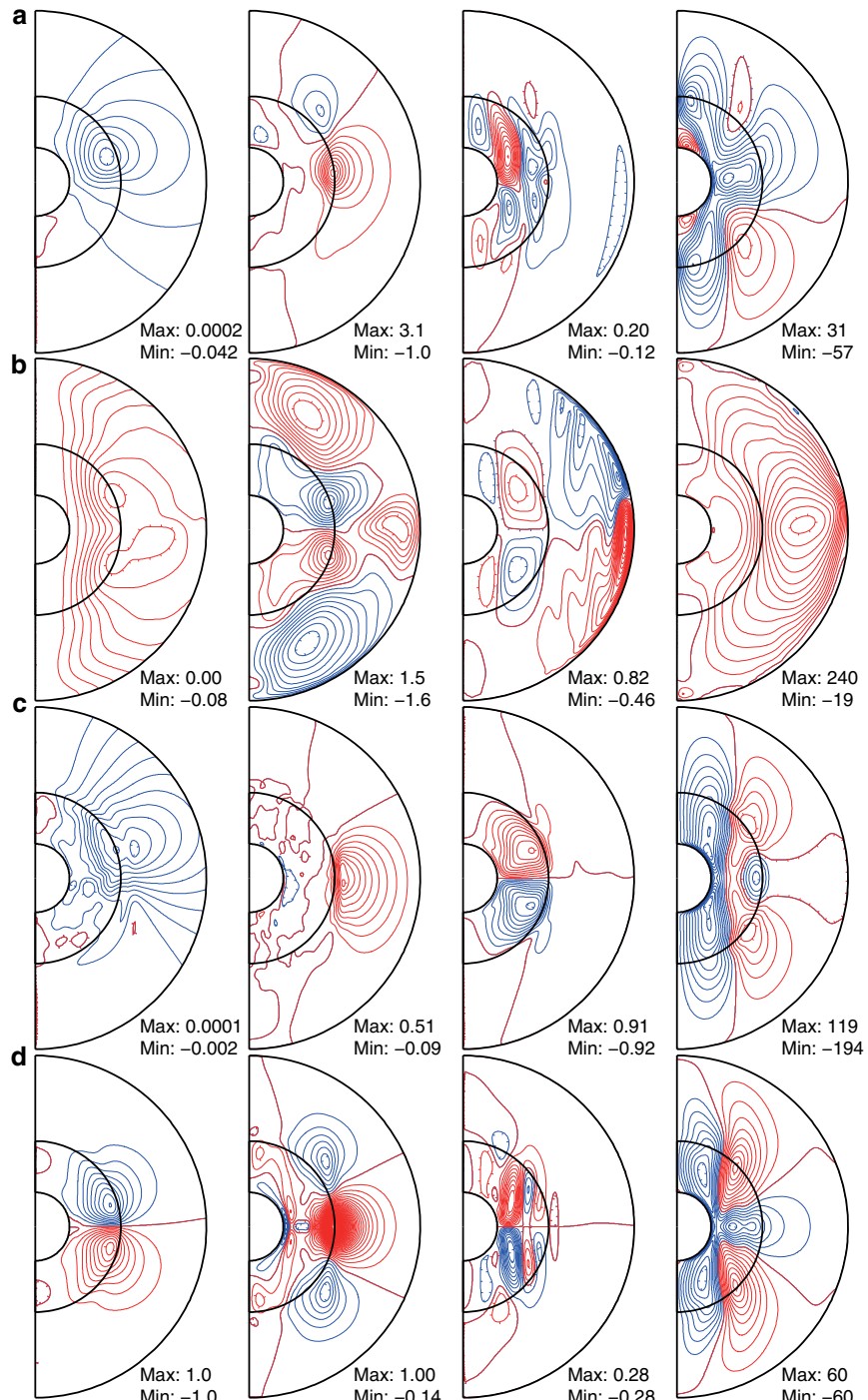

**Fig. 5** Time-averaged axisymmetric structures of non-dimensional magnetic and velocity fields in the meridional cross-section. From the left, poloidal magnetic field lines, toroidal magnetic field, meridional circulation, and zonal flow are drawn, respectively, by contour lines. Red (blue) lines represent positive (negative) values. **a** BU1, **b** TD1, **c** SL1 and **d** BU1K. In **d**, plots of the magnetic field are normalized by the maximum values

imposed by adding a subadiabatic region to $dT_o/dr$ as in previous studies[38,39]:

$$\frac{dT_o}{dr} = \Gamma_o - \frac{1}{2}\left\{1 - \tanh\left(\frac{r - r_s}{d_s}\right)\right\}\left\{\Gamma_o - \frac{r^3 - r_i^3}{r_o^3 - r_i^3}\left(\frac{r_o}{r}\right)^2\right\}, \quad (2)$$

where $r_s$ is the position of the stratification boundary, $d_s$ is the thickness of the transition between convecting and stably stratified regions, and $\Gamma_0$ is the temperature gradient across the stratified layer. Here, we mostly use $r_s = 0.5r_o$, $d_s = 0.05r_o$ and $\Gamma_0 = 10$. Similarly, the radial profile of light element concentration $dC_o/dr$ is given by

$$\frac{dC_o}{dr} = -\frac{\varepsilon_C}{3}r + \frac{a_C}{r^2} = \frac{r^3 - r_o^3}{r_o^3 - r_i^3}\left(\frac{r_i}{r}\right)^2, \quad (3)$$

where $\varepsilon_C = -3r_i^2/(r_o^3 - r_i^3) = -0.097$ and $a_C = -r_i^2 r_o^3/(r_o^3 - r_i^3) = -0.063$. An iron snow layer is represented by superimposing a Gaussian profile onto the basic one as follows:

$$\frac{dC_o}{dr} = \frac{r^3 - r_o^3}{r_o^3 - r_i^3}\left(\frac{r_i}{r}\right)^2 + \Gamma_o \exp\left\{-\frac{(r - r_s)^2}{2d_s^2}\right\}. \quad (4)$$

The radial profiles of non-dimensional heat/compositional flux corresponding to the BU, TD, and SL models summarized in Supplementary Table 1 are shown in Supplementary Fig. 1. The present treatment enables us to examine the effects of different modes of compositional convection while preserving the mode of thermal convection, which is an advantage over the co-density models. These profiles are used to implement a stably stratified layer in a mathematically convenient way,

because the radial profiles of temperature and composition in Mercury's core are still poorly constrained. Those used in a previous study[11] are shown in Supplementary Fig. 4 for comparison.

At both boundaries, the boundary condition for the velocity field is no-slip and insulating for the magnetic field. The influence of treating the inner core as an insulator on the results is insignificant because of its small size[40]. At the outer boundary, we adopted a zero-flux condition for composition and fixed-flux condition for temperature, whereas we assume flux is fixed for composition and either fixed or zero for temperature at the inner boundary. The zero-heat-flux condition is given so as to minimize effects of bottom heating or maximize effects of volumetric internal heating to generate the asymmetric magnetic field[9].

Initial conditions are given by either random perturbations of the temperature and composition, and an axial dipole field as a seed field or the final result of a run at different parameters. The numerical code used in this study is an extended version of refs. [17],[41]. The spatial resolution is 100 or 128 grid points in the radial direction. A spherical harmonics expansion in the angular directions is used up to degree and order 127.

**Axial dipole offset**. Using the Gauss coefficients $(g_L^M, h_L^M)$ up to degree and order two, the eccentricity of the best-fitting dipole from the Hermean center $(x_0, y_0, z_0)$ is given as[42]

$$x_0 = \frac{R_H(L_1 - g_1^1 E)}{3M^2}, y_0 = \frac{R_H(L_2 - h_1^1 E)}{3M^2}, z_0 = \frac{R_H(L_0 - g_1^0 E)}{3M^2},\quad(5)$$

where

$$L_0 = 2g_2^0 g_2^0 + \sqrt{3}(g_1^1 g_2^1 + h_1^1 h_2^1),\quad(6)$$

$$L_1 = -g_1^1 g_2^0 + \sqrt{3}(g_1^0 g_2^1 + g_1^1 g_2^2 + h_1^1 h_2^2),\quad(7)$$

$$L_2 = -h_1^1 g_2^0 + \sqrt{3}(g_1^0 h_2^1 - h_1^1 g_2^2 + g_1^1 h_2^2),\quad(8)$$

$$E = \frac{L_0 g_1^0 + L_1 g_1^1 + L_2 h_1^1}{4M^2},\quad(9)$$

$$M^2 = (g_1^0)^2 + (g_1^1)^2 + (h_1^1)^2.\quad(10)$$

Ignoring non-axisymmetric terms such as Mercury's field, we have $(x_0, y_0, z_0) \sim (0, 0, \frac{R_H g_2^0}{2g_1^0})$. This expression indicates that dipole offset in the axial direction is determined by the ratio of the Hermean-centered axial quadrupole to the centered axial dipole, and also that the offset direction is northward, if these two terms have the same polarity (and southward otherwise).

**Diagnostic quantities**. Important diagnostic quantities of dynamo simulations used in this study include the magnetic Reynolds number, $Rm = Du/\eta$, Elsasser number, $\Lambda = B^2/(2\rho\mu\eta\Omega)$, Gauss coefficient of the axial dipole, $g_1^0$, axial dipole offset normalized by Hermean radius, $D_{offs} = z_0/R_H$, dipole tilt angle, Tilt, dipolarity, $F_{dip}$, fraction of the axisymmetric magnetic field, $F_{axs}$, fraction of the equatorially antisymmetric flow components in the total kinetic energy, $K_{asym}$, local Rossby number $Ro_l$, and the hemispherically-averaged relative axial helicity in each hemisphere, $RAH_{N/S}$, defined as

$$RAH_{N/S} = \frac{\int_{N/S} u_z \omega_z dV}{\sqrt{\int_{N/S} u_z^2 dV}\sqrt{\int_{N/S} \omega_z^2 dV}},\quad(11)$$

where $u_z$ and $\omega_z$ are the axial components of the velocity and vorticity. Volume integration is taken in terms of either the northern (N) or southern (S) hemisphere. The magnetic Reynolds number and Elsasser number are calculated using the root-mean-square values of the flow and magnetic field over the spherical shell. $F_{dip}$ and $F_{axs}$ are calculated at the planetary surface up to spherical harmonic degree four according to MESSENGER observation[2]. These quantities and other input non-dimensional parameters are summarized in Supplementary Table 1. The SL model cannot be a self-sustaining dynamo when the magnetic Reynolds number is not large enough[16]. Based on observations in orbit, we have $g_1^0 = -190$ nT, $D_{offs} = 0.2$, Tilt < 0.8°, $F_{dip} = 79\%$, and $F_{axs} > 99.9\%$ for Mercury[2].

**Helicity analysis**. In order to examine the major contribution to the asymmetric relative axial helicity, let the axial components of the velocity and vorticity be decomposed into two basic modes, the equatorially symmetric mode $(u_z^s, \omega_z^s)$ and antisymmetric mode $(u_z^A, \omega_z^A)$, as follows:

$$u_z = u_z^s + u_z^A,\quad(12)$$

$$\omega_z = \omega_z^s + \omega_z^A.\quad(13)$$

Usually $|u_z^s| \gg |u_z^A|$ and $|\omega_z^A| \gg |\omega_z^s|$ due to the predominance of columnar-style convection in a rotating spherical system. The relative axial helicity in the northern hemisphere $RAH_N$ is then represented as

$$RAH_N = \frac{\int_N (u_z^s \omega_z^A + u_z^A \omega_z^s + u_z^A \omega_z^A + u_z^s \omega_z^s) dV}{\sqrt{\int_N u_z^2 dV}\sqrt{\int_N \omega_z^2 dV}}.\quad(14)$$

The first and second terms in the integrand represent contributions from the flows of the symmetric mode and antisymmetric mode. The remaining two terms are contributions from the interaction of the flows with different symmetry. It is noted that helicity due to a basic mode changes its sign about reflection with respect to the equator, whereas interaction of the basic modes yields an invariant helicity with respect to reflection. Hence, the relative axial helicity in the southern hemisphere can be partly rewritten using the flow components in the northern hemisphere as

$$RAH_S = -\frac{\int_N (u_z^s \omega_z^A + u_z^A \omega_z^s - u_z^A \omega_z^A - u_z^s \omega_z^s) dV}{\sqrt{\int_S u_z^2 dV}\sqrt{\int_S \omega_z^2 dV}}.\quad(15)$$

These expressions suggest that the difference of relative axial helicity in its absolute value is caused by the interaction of the symmetric and antisymmetric flow components when the correlation between the different modes of the velocity $(u_z^s, u_z^A)$ and the vorticity $(\omega_z^S, \omega_z^A)$ is not good. Supplementary Fig. 2 clearly shows that north–south asymmetric helicity distribution in the Mercury-like offset dipolar dynamo can be explained by the evident interaction terms (Supplementary Fig. 2a, b), while the nearly perfect symmetric helicity distributions in the models without dipole offset are due to the velocity field with negligible interaction (Supplementary Fig. 2c–f). We therefore conclude from this analysis that asymmetric helicity due to interaction between different flow modes leads to hemispherically biased dynamo action, and consequently a Mercury-like offset dipolar magnetic field is generated.

We then investigate what mechanisms are responsible for an interaction among flow components that maintained the biased helicity distribution. For this purpose, we carry out three additional runs building on the BU1, where simulations are restarted using the final result of BU1 with some modifications. The first run, BU1L, is restarted with the antisymmetric mode of the velocity, temperature, and composition fields reversed with respect to the equator so that RAH is artificially concentrated in the southern hemisphere rather than in the north. The second run, BU1K, is a kinematic dynamo run, where the Lorentz force term is dropped from the momentum equation. The third, BU1M, is a run of the magnetohydrodynamic dynamo resumed with the symmetric components of the magnetic field reversed in terms of the equator. The first run is designed to see effects of the linear and non-linear hydrodynamic terms and the stably stratified layer, while the second and third runs are intended to show the non-linear effects of the dynamo-generated magnetic field on the core flow. Figure 4 in the main text and Supplementary Fig. 3 clearly show that the helicity bias is sustained by the Lorentz force due to the dynamo action itself; that is, the dipole offset is a natural result of the self-regulation process of the core dynamo prompting symmetry-breaking interaction.

**Magnetic field generation**. In order to investigate the mechanisms of magnetic field generation, we consider the equation for magnetic energy variation:

$$\frac{\partial}{\partial t}\left(\frac{B^2}{2}\right) = \frac{1}{Pm}\mathbf{B}\cdot\nabla^2\mathbf{B} - \mathbf{B}\cdot(\mathbf{u}\cdot\nabla\mathbf{B}) + \mathbf{B}\cdot(\mathbf{B}\cdot\nabla\mathbf{u}).\quad(16)$$

The rightmost term of the equation represents the magnetic energy enhancement achieved by stretching the magnetic field lines due to flow gradient. The stretching terms responsible for generation of the axial dipole, $\mathbf{B}_{P_1}^0$, axial quadrupole, $\mathbf{B}_{P_2}^0$, and axisymmetric toroidal components, $\mathbf{B}_T^0$, are examined by calculating $\mathbf{B}_{P_1}^0 \cdot (\mathbf{B}\cdot\nabla\mathbf{u})$, $\mathbf{B}_{P_2}^0 \cdot (\mathbf{B}\cdot\nabla\mathbf{u})$ and $\mathbf{B}_T^0 \cdot (\mathbf{B}\cdot\nabla\mathbf{u})$, respectively. Time-averaged results are shown in Supplementary Fig. 10.

**Code availability**. Numerical code for dynamo simulations is available upon request.

## Data availability
All relevant data are available from the authors upon request.

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

## Acknowledgements

F.T. was supported by JSPS KAKENHI Grant Numbers JP15K05270, JP15H05834, and JP18K03808. The computation was mainly carried out using the computer facilities at the Research Institute for Information Technology, Kyushu University.

## Author contributions

F.T. performed numerical simulations, data analysis, and manuscript preparation. H.S. and H.T. were involved in project planning, data analysis, and manuscript preparation. All authors contributed to discussion and conclusions of the study.

## Additional information

**Competing interests:** The authors declare no competing interests.

