## [Peer Review File · Nature Communications]

Reviewers' comments:

Reviewer #1 (Remarks to the Author):

Mercury's global magnetic field is special in comparison to the field of other planets in the solar system in three respects: it is unusually weak, highly axisymmetric, and it shows a significant offset of the best-fitting dipole from the center of the planet (or, equivalently, a high ratio axis quadrupole to axial dipole). Several numerical dynamo models are presented in the paper which differ in their distribution of buoyancy sources that drive convection in Mercury's core. The magnetic field in at least one of these models fits, for the first time simultaneously, all three known properties of Mercury's magnetic field. In particular, the authors show that a slight symmetry-breaking between flow in the two hemispheres is facilitated by the feedback of the Lorentz forces on the flow. This is new and interesting. The paper is definitely worth to be published. Some assumptions of the models need to be better justified, or at least their validity for Mercury must be discussed.

Main point:

In the context of these models, two conditions seem to be essential to obtain a magnetic field that matches all properties of Mercury's field: a thick thermally stable layer in the upper part of the core and driving of convection by internal heat sources (the latter requires a significant concentration of radioactive elements, such as uranium, in the core, which is a bit speculative, but possible). I have some problem to envisage a scenario in which the two conditions are mutually compatible. In the case of heating from below, the coexistence of a deep convectively unstable layer and a shallower stable layer is plausible, because heat flux density drops as $1/r^2$ while the conductive adiabatic heat flux rises proportional to r (when one assumes for simplicity constant thermodynamic properties and gravity proportional to r). Hence there can be a cross-over between the two, which marks the boundary between unstable and stable layer. However, if the heating is by homogeneously distributed internal sources, also the heat flux varies proportional to r . Then there is no cross-over and the entire fluid core is either stable or unstable. In addition, setting the heat flow at the inner-core boundary to zero is not compatible with having at same time a compositional flux. Compositional flux depends on inner-core growth, which also implies a significant release of latent heat. Perhaps some plausible arguments can be found to justify the assumptions, e.g. that the inner-core heat flow takes just the value of the adiabatic conductive heat (which must be deducted in a Boussinesq model), and that for some reason the radioactive elements are concentrated only in the deep layer. However, I do not see any obvious justification for the latter assumption.

Another possibility could be that, for homogeneous internal heating in the core, the adiabatic temperature gradient varies super-linearly with the radius because of strong pressure-dependence of the thermal expansion coefficient, so that the adiabatic conductive flux can exceed the actual heat flux in the upper parts of the core, but falls short of it in the deep parts. This would need verification on the basis of the available data on the thermodynamic properties of iron and structural models for Mercury.

Minor comments:

- The use of the term "feedback dynamo" in the title may lead to a misunderstanding. It has been used for a model invoking the feedback of the magnetospheric field on the dynamo in the core (Heyner et al., Science, 334, 1690, 2011).
- The wording and the English in general could be improved in several places. For example, using the term "new paradigm" in the first sentence of the paper is not appropriate.
- Why is in Fig.2 not the result for model BU2 shown, which gives a better fit to the observed dipole offset? Is in this case the agreement in the amplitude of the field less satisfactory? Because field intensity is one of the criteria to judge Mercury dynamo models, some measure of it (e.g. the g_1^0 Gauss coefficient) should be added to supplementary table S1.

- The authors emphasize that the codensity-formulation may be inadequate and use different diffusivities for temperature and composition. But it is not shown that, in their models, double-diffusive effects play an essential role. For example, is there any indication of fingering-type instability in the thermally stable layer, as found in ref. 14 ? If not, perhaps the degree of thermal stability in the upper layer is so overwhelming in comparison to the degree of compositional instability that no significant difference exists compared to an equivalent codensity model?
- Concerning Mercury's observed field, it would be worthwhile to refer also to the paper by Thebault et al. (Phys. Earth Planet. Int., 276, 93, 2018) in addition to the papers by Anderson et al. Using a larger set of data, the latter paper finds similar results, but they prefer a slightly weaker quadrupole-to-dipole ratio.
- Ref. 12 is quoted for the value of magnetic diffusivity (in the caption of Fig. 2). This is not adequate because that paper estimates conductivity values at the pressures and temperatures of the Earth's core, which are much larger than those in Mercury's core.
- Figure caption 3: A figure 1b is mentioned, but there is no Fig. 1b; "rotation is aligned with the z-axis", but no z-axis is shown in the figure.
- Figure 4a: I assume that the clouds represent a large number of snapshots at different times. This should be mentioned in the caption.

Reviewer #2 (Remarks to the Author):

The authors have found a numerical dynamo model that produces a magnetic field that resembles the field of Mercury, in that it is more active in one hemisphere, leading to an apparent offset of the centre of the dipole.

What's new here, is that an apparently stable dipole field is produced using spherically symmetric boundary conditions.

It is a novel and interesting finding, which deserves publication.

Furthermore, there is a good match with some of the properties of the Hermean field.

My main criticism is that the authors give no good explanation for this good match.

Is it pure luck?

What makes their model special? How does the double-diffusive setup help to produce a hemispherical dynamo?

Why does BU work and not SL or TD? Are the flows very different?

Because of this lack of explanation (the relative helicity analysis just pushes the mystery from magnetic field asymmetry to helicity asymmetry),

we cannot be certain that this rather high viscosity study (both in terms of Ekman and magnetic Prandtl numbers) can be extrapolated

to Mercury's conditions.

If the authors could (i) reproduce the same features at lower viscosity (dropping Ekman number to $E=1e-5$ say,

and setting the magnetic Prandtl number so that the Rm is low) or (ii) give a physical explanation linking the features of BU to the asymmetric field,

it would make their case much stronger.

I suggest the authors revise their manuscript to address these comments, as well as the remarks and questions below.

Otherwise, the supplementary material gives important information for specialists and allows to reproduce the work.

Nathanael Schaeffer.

Other important remarks:

1) In the introduction about other hemispherical dynamos, you should compare to other hemispherical dynamos and tell why yours is different.

Important references to discuss here are:

- with homogeneous boundary conditions:

* Grote & Busse 2000 <https://doi.org/10.1103/PhysRevE.62.4457>

* Gallet & Petrelis 2009 <https://doi.org/10.1103/PhysRevE.80.035302>

- and (to a lesser extent) with heterogeneous boundary conditions:

* Stanley+ 2008 <https://doi.org/10.1126/science.1161119>

* Amit+ 2011 <https://doi.org/10.1016/j.pepi.2011.07.008>

* Dietrich & Wicht 2013 <https://doi.org/10.1016/j.pepi.2013.01.001>

2) It should also be acknowledged in the text that so far the measurements made about Mercury's field are very biased towards the northern hemisphere

(the probe's orbit is taking it too far from the planet in the southern hemisphere).

It is not impossible that the dipole offset is an artefact of these measurements.

2) Only mean values are given for the various diagnostics (both in the text line 77, and in table S1). Are these values stable? What is the standard deviation?

What about showing an histogram, most importantly for the displacement of the magnetic equator, D_{off} ?

Similarly, in Fig 2, is it snapshots that are represented? Or time-averages? (how representative?)

3) from line 90 to line 103, all this discussion about Helicity should be removed from the main text and left only as supplementary material.

It distracts the reader from the main message and is even not a satisfactory explanation: What in your BU2 setup is so special to allow this asymmetry? The "interaction" of different flow modes? What makes double-diffusive convection special about it? Those question are left unanswered by the helicity analysis.

However, what is interesting and should be kept in the text, is the fact that the magnetic field itself sustains the flow asymmetry.

(Although, again, why is BU2 so special about it ?).

For a detailed feedback mechanism of magnetic field on helicity, please refer to Sreenivasan+ 2011 <https://doi.org/10.1017/jfm.2011.233>

Your findings may well be related to subcritical dynamo action.

I also wonder if the initial profiles (shown in fig S1) are significantly altered in an asymmetric way?

It would be instructive to show these profiles (T_0+T and C_0+C), maybe superimposed to fig S1, in dashed lines (see e.g. fig A2 in Schaeffer+ 2017 <https://doi.org/10.1093/gji/ggx265>).

Some other unanswered questions:

102-103: why is there negligible interaction in TD and SL? Is it an effect of the magnetic field?

Have you tried starting TD and SL from the magnetic field of BU? How different are the flows?

More or less columnar? etc...

4) In table S1, cases BU2 and BU3 lead to very similar R_m , despite the large changes in Rayleigh number. How is this possible?

Typos:

l36 and l64: "the Mercury's" => Mercury's

l99: "flows are interacted" => flows interact

Reviewer #3 (Remarks to the Author):

- Key results:

The authors perform numerical dynamo simulations to investigate Mercury's magnetic field and are able to reproduce the planet's north-south asymmetry, strong axisymmetry, and weak field strength. They argue that magnetic feedback is a necessary component of generating the axially offset dipolar field.

- Originality and significance:

The role of an outer stable layer, double-diffusive convection, and bottom-up versus top-down versus snow layer convective modes have all been studied previously in the context of planetary dynamos, including for Mercury in particular (e.g., Christensen 2006; Christensen and Wicht 2008; Stanley and Mohammadi 2008; Manglik et al. 2010; Vilim et al. 2010; Cao et al. 2014; Tian et al. 2015). While this manuscript considers some new combinations of these elements to reproduce key elements of Mercury's magnetic field (dipole offset, strong axisymmetry, weak field strength) simultaneously for the first time, I am concerned that the study is not sufficiently novel for publication here.

Early studies focused on the weak magnetic field strength, which is often regarded to be due to a thick stably stratified layer in the core (e.g., Christensen 2006; Christensen and Wicht 2008; cf. Glassmeier et al. 2007; Heyner et al. 2011; Vilim et al. 2010). The surface magnetic fields of these models were dominated by the axial dipole component, the axial quadrupole component, or a combination of the two. Moreover, secular variation of these dynamos was low when extrapolated to the planet surface.

The addition of double diffusion to these models causes the magnetic field strength to increase due to flows that are generated in the thermally stratified layer (Manglik et al. 2010), although these authors note that a double diffusive dynamo that matches the observed field strength could likely be found if the input parameters were tuned in their low compositional buoyancy flux case. The manuscript would benefit from a more explicit discussion of how their simulations extend these results (see also later comment). Similarly, the manuscript does not explain if (or why) double-diffusive convection is a necessary ingredient for generating a Mercury-like field or if a co-density approach would likely be adequate.

Cao et al. (2014) considered volumetrically distributed versus bottom-driven convection and concluded that the former is a necessary ingredient for axial-dominant, equatorially asymmetric dynamos. Local excess equatorial CMB heat flow causes these solutions to be steady in time. They further argue that this magnetic field morphology is the result of equatorially asymmetric helicity that results from the mutual excitation of two different modes of columnar convection, similar to this manuscript.

The role of magnetic field feedback on the maintenance of equatorially asymmetric helicity is novel and very interesting. However, this result is not discussed sufficiently (see also later comment). Given the weak magnetic field strengths of the models, why does this feedback occur? Is this phenomena unique to Mercury? If so, why?

- Data, methodology, statistics, and uncertainties:

The thermal and compositional profiles used in these models appear to be a critical component for generating Mercury-like magnetic fields, and are different from those employed in previous models for Mercury's dynamo. These profiles are described in detail, but not justified in a physical way. In particular, the motivation and realism of tanh functions are not explained sufficiently, especially in the context of radial interior structure/temperature/composition estimates.

The manuscript does not show or describe the temporal evolution of the magnetic field in much

detail. For example, how do the Gauss coefficients vary with time and how does this compare to observations? What is the standard deviation of the magnetic spectra shown in Figure 2e-f?

The thickness of the mantle, which is needed to extrapolate the dynamo simulation results to the planet surface, is not specified in the text.

No discussion on the impact of assuming (necessarily) high Ekman and magnetic Prandtl numbers and a low Lewis number was included.

- Conclusions:

The manuscript argues that thermal convection due to internal heating is preferred in Mercury's core and, therefore, that a considerable amount of radioactive heat sources are required. Is this consistent with thermal evolution models and compositional constraints?

The last paragraph is very speculative and necessitates additional simulations with different thicknesses of the stably stratified layer.

- Suggested improvements:

I recommend addressing the temporal evolution of the magnetic fields obtained in more detail. A discussion of dipole tilt angle in addition to F_{ax} would enable further comparison with observations and the broader literature. Inclusion of the local Rossby number (e.g., Christensen and Aubert, GJI, 2006) would also help characterize the flow.

It would be helpful to further distinguish the differences between these models and those of Manglik et al. (2010). The radial heat and composition flux profiles are very different, and this should be highlighted more since the other input parameters are quite similar. For example, adding panels to Figure S1 for the high-sulphur and low-sulphur cases from Manglik et al. (2010), with similar non-dimensionalizations, would enable a better comparison between the models.

Similarly, the authors could better address why magnetic field feedback on the flow is necessary to generate the helicity asymmetry. Sreenivasan et al. (2014), Soderlund et al. (2015), and Aurnou and King (2017) may be helpful in this regard.

Sreenivasan, B., Sahoo, S., & Dhama, G. (2014). The role of buoyancy in polarity reversals of the geodynamo. *Geophysical Journal International*, 199(3), 1698-1708.

Soderlund, K. M., Sheyko, A., King, E. M., & Aurnou, J. M. (2015). The competition between Lorentz and Coriolis forces in planetary dynamos. *Progress in Earth and Planetary Science*, 2(1), 24.

Aurnou, J. M., & King, E. M. (2017). The cross-over to magnetostrophic convection in planetary dynamo systems. In *Proc. R. Soc. A* 473(2199), 20160731. The Royal Society.

- References:

Knibbe and van Westrenen (2018) would likely be of interest to the authors since it discusses the thermal and magnetic evolution of Mercury assuming an Fe-Si core and implements a conductive temperature profile in the core:

Knibbe, J. S., & van Westrenen, W. (2018). The thermal evolution of Mercury's Fe-Si core. *Earth and Planetary Science Letters*, 482, 147-159.

Landeau and Aubert (2011) find hemispheric dynamos with equatorially asymmetric convection

and would be appropriate to cite:

Landeau, M., & Aubert, J. (2011). Equatorially asymmetric convection inducing a hemispherical magnetic field in rotating spheres and implications for the past martian dynamo. *Physics of the Earth and planetary interiors*, 185(3-4), 61-73.

- Clarity and context:

The manuscript is not particularly well written. The models are not clearly described in the main text; it would be helpful to at least state how many models were run and what's varied between them.

Replies to Reviewer #1

First of all, the authors would like to thank the reviewer for her/his valuable comments, which undoubtedly help us improve the manuscript.

In the following, reviewer's comments are written in bold italic.

Mercury's global magnetic field is special in comparison to the field of other planets in the solar system in three respects: it is unusually weak, highly axisymmetric, and it shows a significant offset of the best-fitting dipole from the center of the planet (or, equivalently, a high ratio axis quadrupole to axial dipole). Several numerical dynamo models are presented in the paper which differ in their distribution of buoyancy sources that drive convection in Mercury's core. The magnetic field in at least one of these models fits, for the first time simultaneously, all three known properties of Mercury's magnetic field. In particular, the authors show that a slight symmetry-breaking between flow in the two hemispheres is facilitated by the feedback of the Lorentz forces on the flow. This is new and interesting. The paper is definitely worth to be published. Some assumptions of the models need to be better justified, or at least their validity for Mercury must be discussed.

We thank the reviewer for her/his positive and constructive review. We hope our point-by-point responses given below will be satisfactory to the reviewer.

Main point:

Comment 1:

In the context of these models, two conditions seem to be essential to obtain a magnetic field that matches all properties of Mercury's field: a thick thermally stable layer in the upper part of the core and driving of convection by internal heat sources (the latter requires a significant concentration of radioactive elements, such as uranium, in the core, which is a bit speculative, but possible). I have some problem to envisage a scenario in which the two conditions are mutually compatible. In the case of heating from below, the coexistence of a deep convectively unstable layer and a shallower stable layer is plausible, because heat flux density drops as $1/r^2$ while the conductive adiabatic heat flux rises proportional to r (when one assumes for simplicity constant thermodynamic properties and gravity proportional to r). Hence there can be a cross-over between the two, which marks the boundary between unstable and stable layer. However, if the heating is by homogeneously distributed internal

sources, also the heat flux varies proportional to r . Then there is no cross-over and the entire fluid core is either stable or unstable. In addition, setting the heat flow at the inner-core boundary to zero is not compatible with having at same time a compositional flux. Compositional flux depends on inner-core growth, which also implies a significant release of latent heat. Perhaps some plausible arguments can be found to justify the assumptions, e.g. that the inner-core heat flow takes just the value of the adiabatic conductive heat (which must be deducted in a Boussinesq model), and that for some reason the radioactive elements are concentrated only in the deep layer. However, I do not see any obvious justification for the latter assumption.

Another possibility could be that, for homogeneous internal heating in the core, the adiabatic temperature gradient varies super-linearly with the radius because of strong pressure-dependence of the thermal expansion coefficient, so that the adiabatic conductive flux can exceed the actual heat flux in the upper parts of the core, but falls short of it in the deep parts. This would need verification on the basis of the available data on the thermodynamic properties of iron and structural models for Mercury.

Reply:

Strictly speaking, thermodynamic consistency should be examined for the radial profile with possible composition and property of material in Mercury's core. However, because they are still poorly constrained, it is worth assuming radial temperature profiles and composition in a mathematically convenient way and running a number of dynamo calculations to find essential elements to produce magnetic field that has similar character to the Mercury's field. We considered that stratification and buoyancy forcing in the core are essential elements to determine the dynamo. The profile used in this paper has a property that the reference thermal structure outside the stably stratified layer is not seriously affected by that within the layer. This formulation allows us to easily see effects of the stably stratified layer with different thickness on convection and resultantly the magnetic field. Some words about thickness of the Mercury's stably stratified layer were added in the last paragraph of the revised manuscript.

The assumption for the buoyancy forcing is considered to be a kind of an end-member model to maximize effects of internal heating as described in the manuscript. Concerning zero heat flux condition at the inner core boundary, we have implicitly supposed that the inner core heat flux is very close to the adiabatic one, same as your first suggestion. Although it is not included in the manuscript, we also conjecture that the radioactive elements in the fluid core such as Uranium are not included in the inner core and stay in the outer core, because it is incompatible with solid phase. We consider zero heat flux condition to be a first approximation to such a situation at ICB.

We added some description and discussion in the revised manuscript (lines 184-188), and Supplementary Information (page 2) regarding the issues.

We began with the model as a first step to solve a mystery of Mercury's dynamo. In the future, we may be able to estimate thickness of the stably stratified layer in Mercury's core by combining numerical modeling and results of coming magnetic field observations.

Minor comments:

Comment 2:

- The use of the term "feedback dynamo" in the title may lead to a misunderstanding. It has been used for a model invoking the feedback of the magnetospheric field on the dynamo in the core (Heyner et al., Science, 334, 1690, 2011).

Reply:

According to the reviewer's suggestion, we changed the term in the title from "feedback dynamo" to "self-regulating dynamo".

Comment 3:

- The wording and the English in general could be improved in several places. For example, using the term "new paradigm" in the first sentence of the paper is not appropriate.

Reply:

Following the suggestion, we carefully checked English. The term "new paradigm" was replaced by "new regime". Thanks.

Comment 4:

- Why is in Fig.2 not the result for model BU2 shown, which gives a better fit to the observed dipole offset? Is in this case the agreement in the amplitude of the field less satisfactory? Because field intensity is one of the criteria to judge Mercury dynamo models, some measure of it (e.g. the g_1^0 Gauss coefficient) should be added to supplementary table S1

Reply:

As a criterion, we added mean dipole tilt angle to Table S1. The dipole tilt angle in BU1 is smaller than that in BU2, and therefore, more similar to the Mercury's magnetic field. Also as in your comment 6 below, the latest estimate by Thebault et al. (2018) shows a quadrupole-to-dipole ratio slightly smaller than that by Anderson et al., and good match for BU1. Based on their estimate and a smaller dipole tilt angle, we have kept BU1 to be shown in

Fig. 2.

Comment 5:

- The authors emphasize that the codensity-formulation may be inadequate and use different diffusivities for temperature and composition. But it is not shown that, in their models, double-diffusive effects play an essential role. For example, is there any indication of fingering-type instability in the thermally stable layer, as found in ref. 14 ? If not, perhaps the degree of thermal stability in the upper layer is so overwhelming in comparison to the degree of compositional instability that no significant difference exists compared to an equivalent codensity model?

Reply:

In some runs, fingering-type convection is found in the thermally stably stratified layer. We added typical flow structure in Fig S5. In such cases, magnetic field morphology is not Mercury-like as in ref. 14. Rather than that, the main point of considering different diffusivities is what enables us to explicitly discern buoyancy source distributions due to temperature and composition with different diffusivity: thermal convection by volumetric heating, which is an important ingredient (Cao et al., 2014), and compositional convection due to inner core growth. This cannot be done in the co-density model. We think this treatment is essential in the present study. Some sentences regarding the issue were added for discussion in the revised version (lines 158-174).

Comment 6:

- Concerning Mercury's observed field, it would be worthwhile to refer also to the paper by Thebault et al. (Phys. Earth Planet. Int., 276, 93, 2018) in addition to the papers by Anderson et al. Using a larger set of data, the latter paper finds similar results, but they prefer a slightly weaker quadrupole-to-dipole ratio.

Reply:

According to the reviewer's suggestion and also in relation to comment 4, Thebault et al. was cited in the revised version. Thank you for pointing it out.

Comment 7:

- Ref. 12 is quoted for the value of magnetic diffusivity (in the caption of Fig. 2). This is not adequate because that paper estimates conductivity values at the pressures and temperatures of the Earth's core, which are much larger than those in Mercury's core.

Reply:

The value of magnetic diffusivity was changed to 1 as a typical value. Correspondingly, Gubbins and Roberts (1987) in *Geomagnetism Vol. 2* was cited instead of the reference in the original version. Accordingly, magnetic field scale was slightly changed.

Comment 8:

- Figure caption 3: A figure 1b is mentioned, but there is no Fig. 1b; "rotation is aligned with the z-axis", but no z-axis is shown in the figure.

Reply:

This is a typo for figure 2b, which was corrected in the revised version. The z-axis was added to figure 3. Thank you for pointing them out.

Comment 9:

- Figure 4a: I assume that the clouds represent a large number of snapshots at different times. This should be mentioned in the caption.

Reply:

Your interpretation is correct. Some words were added to the caption in the revised version to make it clearer.

Other changes

Figure S1f was replaced, because the profiles for SL2 were not drawn in the original version.

Replies to Reviewer #2

First of all, the authors would like to thank the reviewer (Dr. N. Schaeffer) for his valuable comments, which undoubtedly help us improve the manuscript.

In the following, reviewer's comments are written in bold italic.

Comment 1:

My main criticism is that the authors give no good explanation for this good match.

Is it pure luck?

What makes their model special? How does the double-diffusive setup help to produce a hemispherical dynamo?

Why does BU work and not SL or TD? Are the flows very different?

Because of this lack of explanation (the relative helicity analysis just pushes the mystery from magnetic field asymmetry to helicity asymmetry), we cannot be certain that this rather high viscosity study (both in terms of Ekman and magnetic Prandtl numbers) can be extrapolated to Mercury's conditions.

If the authors could (i) reproduce the same features at lower viscosity (dropping Ekman number to $E=1e-5$ say, and setting the magnetic Prandtl number so that the Rm is low) or (ii) give a physical explanation linking the features of BU to the asymmetric field, it would make their case much stronger.

I suggest the authors revise their manuscript to address these comments, as well as the remarks and questions below.

Otherwise, the supplementary material gives important information for specialists and allows to reproduce the work.

Reply:

The findings of the present manuscript are obtained from our broad survey for dynamos driven by double diffusive convection due to various types of buoyancy source distribution. Using double diffusive setup enables us to distinguish contributions of temperature and composition to such buoyancy source distributions, which cannot be done in the widely used co-density framework. This is a point of our model. Also, it is already known in one of our previous studies that double diffusive convection can yield dynamos different from those by the corresponding co-density (Takahashi, 2014 PEPI).

Actually, typical flow structures are fingering-type convection in TD like Manglik et al. (2010), and two-layered convection in SL like Vilim et al. (2010). Flow structures in runs of BU, TD

and SL were included in Supplementary Information (Fig. S5). Also, the radial rms-velocity and magnetic field profiles were shown in Fig S6, where it is found that characteristic structures are different among these models. It is noted that the axisymmetric toroidal magnetic field around the stratification boundary is prominent in BU. Such a strong toroidal field generation is explained by the omega-effect due to zonal wind around the stably stratified layer. The strong zonal toroidal field would act on the flow to enlarge the azimuthal flow scale even at a lower Ekman number (Takahashi and Shimizu, 2012 JFM; Matsui et al., 2014 G-cubed). We expect the present results to be applicable at more Mercury-like parameters as long as typical flow and magnetic field structures like Fig. S6a are established, although we admit that more extensive parameter survey about Ekman number and magnetic Prandtl number is needed

Deciphering the mechanisms for helicity asymmetry could give us insights into those for the magnetic field asymmetry. We are considering the MHD, where the magnetic field could have non-linearly an impact on the velocity field, and the present results are obviously the case. Actually, it is understood by helicity analysis and a kinematic dynamo run how a hemispherical dynamo is achieved and maintained in case of BU. The hemispherical magnetic field is generated by asymmetric helicity arising from non-linear interaction between the symmetric and antisymmetric flows, and at the same time, prompts non-linear interaction between these flows yielding the asymmetric helicity.

In the kinematic run without the effects of the Lorenz force on the flows, convection restores almost perfect equatorial symmetry, and the resultant dynamo is quadrupolar. The kinematic result indicates that the antisymmetric flow component is maintained by the magnetic field (see newly added Fig. S8 exhibiting rapid decay of the antisymmetric flows), and the magnetically-driven, antisymmetric flow interacts with the symmetric flow to generate the asymmetric helicity, and resultantly the axial dipole comparable to the axial quadrupole. It is also suggested that the antisymmetric flow components in other runs of TD and SL failing to have the dipole offset even with a similar fraction in kinetic energy (K_{asym}) are not magnetically-driven. Taking them and the fact that even slight asymmetry has a substantial impact on the resultant field morphology (Gallet and Petrelis, 2009) into account, we could infer the reason why BU runs yield the asymmetric dynamo as (1) the flows work to have a quadrupolar morphology in a kinematic dynamo; (2) the antisymmetric flow components are maintained by the magnetic field in an MHD dynamo. At present, however, it is not clear that why the BU flows without magnetic feedback prefer the quadrupolar dynamo. This issue remains to be examined in future works. Kinematic dynamo study could be helpful in this regard. We added the above issue as discussion in the revised version (lines 138-157).

Other important remarks:

Comment 2:

1) In the introduction about other hemispherical dynamos, you should compare to other hemispherical dynamos and tell why yours is different.

Important references to discuss here are:

- with homogeneous boundary conditions:

**** Grote & Busse 2000 <https://doi.org/10.1103/PhysRevE.62.4457>***

**** Gallet & Petrelis 2009 <https://doi.org/10.1103/PhysRevE.80.035302>***

- and (to a lesser extent) with heterogeneous boundary conditions:

**** Stanley+ 2008 <https://doi.org/10.1126/science.1161119>***

**** Amit+ 2011 <https://doi.org/10.1016/j.pepi.2011.07.008>***

**** Dietrich & Wicht 2013 <https://doi.org/10.1016/j.pepi.2013.01.001>***

Reply:

We are grateful to the reviewer for suggesting important literatures. In particular, Gallet & Petrelis (2009) shows important results to interpret our results in terms of non-linear symmetry breaking interaction, although their results rely on linear calculations of α^2 -dynamo. We added a paragraph for discussion citing these papers in the revised version (lines 158-174).

Comment 3:

2) It should also be acknowledged in the text that so far the measurements made about Mercury's field are very biased towards the northern hemisphere

(the probe's orbit is taking it too far from the planet in the southern hemisphere).

It is not impossible that the dipole offset is an artefact of these measurements.

Reply:

A sentence about the issue was added in the revised manuscript (lines 39-41).

Comment 4:

2) Only mean values are given for the various diagnostics (both in the text line 77, and in table S1). Are these values stable? What is the standard deviation?

What about showing an histogram, most importantly for the displacement of the magnetic equator, Doff?

Similarly, in Fig 2, is it snapshots that are represented? Or time-averages? (how representative?)

Reply:

To show time variation, time-series of the Gauss coefficients for the axial dipole, axial quadrupole, and dipole offset, D_{off} , were added in Fig. S7. As for Fig. 2, error bars at 1-standard-deviation level were added to time-averaged magnetic power spectra. The radial fields are snapshots. The axial dipole and quadrupole vary slowly. Hence, we consider even a snapshot can well show representative features.

Comment 5:

3) from line 90 to line 103, all this discussion about Helicity should be removed from the main text and left only as supplementary material.

It distracts the reader from the main message and is even not a satisfactory explanation: What in your BU2 setup is so special to allow this asymmetry? The "interaction" of different flow modes?

What makes double-diffusive convection special about it? Those question are left unanswered by the helicity analysis.

However, what is interesting and should be kept in the text, is the fact that the magnetic field itself sustains the flow asymmetry.

(Although, again, why is BU2 so special about it ?).

For a detailed feedback mechanism of magnetic field on helicity, please refer to Sreenivasan+2011 <https://doi.org/10.1017/jfm.2011.233>

Your findings may well be related to subcritical dynamo action.

Reply:

We do not agree with the reviewer's suggestion. This part also contributes to the main message of the present manuscript. We are afraid that removing it from the main text would cause readers to miss an important point of the results. Asymmetric helicity production and the magnetic feedback should be considered together to correctly understand the fundamental physical processes for sustenance of the dipole offset, as in reply to your comment 1. Here, we do not repeat the same argument.

According to the reviewer's suggestion, Sreenivasan and Jones (2011) was cited for discussion. In fact, the present result is supercritical, because the magnetic field grows exponentially with time in a run of BU1K. Nevertheless, Sreenivasan and Jones (2011) is worth being cited for discussion. We added some words about supercritical behavior in the caption of Figure 4 in the revised manuscript.

Comment 6:

I also wonder if the initial profiles (shown in fig S1) are significantly altered in an

asymmetric way?

It would be instructive to show these profiles (T_0+T and C_0+C), maybe superimposed to fig S1, in dashed lines (see e.g. fig A2 in Schaeffer+ 2017 <https://doi.org/10.1093/gji/ggx265>).

Reply:

Profiles of heat/compositional flux differences between the values at north and south poles are added as Fig. S9. Although we tried superimposing the profiles of flux themselves to Fig S1, asymmetry amplitude is faint compared with the reference profiles, and could hardly discern differences. Therefore, we show them separately.

Some other unanswered questions:

Comment 7:

I102-103: why is there negligible interaction in TD and SL? Is it an effect of the magnetic field? Have you tried starting TD and SL from the magnetic field of BU? How different are the flows? More or less columnar? etc...

Reply:

Our response to most part of this comment is described in reply to your comment 1. We do not repeat it here. Apart from that, it is noted that the identical initial conditions are used for some BU and TD runs, and then, we have obtained different results.

Comment 8:

4) In table S1, cases BU2 and BU3 lead to very similar R_m , despite the large changes in Rayleigh number. How is this possible?

Reply:

It is due to different basic thermal profiles between BU2 and BU3.

Comment 9:

Typos:

l36 and l64: "the Mercury's" => Mercury's

l99: "flows are interacted" => flows interact

Reply:

Corrected. Thank you for pointing them out.

Other changes

Title is slightly changed according to the comment by reviewer#1.

Figure S1f was replaced, because the profiles for SL2 were not drawn in the original version.

Replies to Reviewer #3

First of all, the authors would like to thank the reviewer for her/his valuable comments, which undoubtedly help us improve the manuscript.

In the following, reviewer's comments are written in bold italic.

- Originality and significance:

Comment 1:

The role of an outer stable layer, double-diffusive convection, and bottom-up versus top-down versus snow layer convective modes have all been studied previously in the context of planetary dynamos, including for Mercury in particular (e.g., Christensen 2006; Christensen and Wicht 2008; Stanley and Mohammadi 2008; Manglik et al. 2010; Vilim et al. 2010; Cao et al. 2014; Tian et al. 2015). While this manuscript considers some new combinations of these elements to reproduce key elements of Mercury's magnetic field (dipole offset, strong axisymmetry, weak field strength) simultaneously for the first time, I am concerned that the study is not sufficiently novel for publication here.

Reply:

We hope our point-by-point responses given below will alleviate the reviewer's concern about our results.

Comment 2:

The addition of double diffusion to these models causes the magnetic field strength to increase due to flows that are generated in the thermally stratified layer (Manglik et al. 2010), although these authors note that a double diffusive dynamo that matches the observed field strength could likely be found if the input parameters were tuned in their low compositional buoyancy flux case. The manuscript would benefit from a more explicit discussion of how their simulations extend these results (see also later comment). Similarly, the manuscript does not explain if (or why) double-diffusive convection is a necessary ingredient for generating a Mercury-like field or if a co-density approach would likely be adequate.

Reply:

Double-diffusive convection modelling enables us to treat thermal and compositional buoyancy separately applying different diffusivity coefficients. Hence, we can distinctly put buoyancy source at the ICB for compositional convection, and within the core for thermal convection. It is

a great advantage of this treatment, whereas the co-density formulation cannot tell the difference in principle. As the reviewer pointed out below, contribution from a volumetrically distributed source seems to be necessary for a Mercury-like dynamo. Hence, we think separation between bottom-driven and internally-driven convection is a key element for the present results. Moreover, considering double-diffusive convection could provide various implications for thermal evolution, and chemical composition of Mercury as well as core dynamics. We added discussion with previous results, and some words for the necessity of double-diffusive treatment in the revised version (lines 158-174).

Comment 3:

Cao et al. (2014) considered volumetrically distributed versus bottom-driven convection and concluded that the former is a necessary ingredient for axial-dominant, equatorially asymmetric dynamos. Local excess equatorial CMB heat flow causes these solutions to be steady in time. They further argue that this magnetic field morphology is the result of equatorially asymmetric helicity that results from the mutual excitation of two different modes of columnar convection, similar to this manuscript.

The role of magnetic field feedback on the maintenance of equatorially asymmetric helicity is novel and very interesting. However, this result is not discussed sufficiently (see also later comment). Given the weak magnetic field strengths of the models, why does this feedback occur? Is this phenomena unique to Mercury? If so, why?

Reply:

We agree with the reviewer's argument about the results by Cao et al. (2014), which is seemingly similar to our results. As the reviewer mentioned, an important point in the present results is that the Lorentz force plays a critical role in yielding and maintaining an equatorially asymmetric helicity distribution. Mechanisms responsible for generating and maintaining helicity asymmetry here is not explained by a linear superposition of the symmetric and antisymmetric velocity fields as in Cao et al. (2014), but by the non-linear interaction between two different modes prompted by feedback effects of the Lorentz force, which is evidently demonstrated by our kinematic run. We consider the magnetic field within the convective region is strong enough for this feedback to work, in particular near the stable and unstable stratification boundary, whereas the weak magnetic field at the surface is due to attenuation associated with skin effects of the stably stratified layer. In order to show them clearly, we added plots of the rms-magnetic field profiles with respect to the radius, and time-series of the kinetic energy in the antisymmetric component in the kinematic run as Figs. S6 and S8.

Based on the present results, it is found that distinct distribution of thermo-compositional

buoyancy source as well as a thick stably stratified layer is important to reproduce Mercury-like morphology and strength of the magnetic field. Although the magnetic feedback should generally occur, how effectively it works may depend on some conditions such as buoyancy source distribution, thickness of the stably stratified layer, inner core size and so on, all of which reflect thermo-chemical evolution of the planets. In this sense, the observable feedback may be specific to Mercury's core. It is very interesting and important to examine difference with a body similar to Mercury in size such as Ganymede, which remains as a future work.

- Data, methodology, statistics, and uncertainties:

Comment 4:

The thermal and compositional profiles used in these models appear to be a critical component for generating Mercury-like magnetic fields, and are different from those employed in previous models for Mercury's dynamo. These profiles are described in detail, but not justified in a physical way. In particular, the motivation and realism of tanh functions are not explained sufficiently, especially in the context of radial interior structure/temperature/composition estimates.

Reply:

Since the radial profiles of temperature and composition in Mercury's core are not well constrained at present as far as we know, we choose the heat flux profile to ensure stable stratification in the upper part of the core. In this sense, using the form of hyperbolic tangent is for mathematical convenience, which allows a continuous change in heat flux between convecting and stable regions as in, for example, Takehiro and Lister (2001). This formulation allows us to put the stably stratified region without seriously affecting the radial profile outside the stable layer (effects are limited within transition thickness given as a parameter), and thus to easily see effects of the stably stratified layer with different thickness on convection and resultantly the magnetic field. In some previous studies, an abrupt, discontinuous change in heat or co-density flux is imposed to implement the stably stratified layer (e.g. Stanley and Mohammadi, PEPI, 2008; Vilm et al., JGR, 2010). The present way is an extension of these corresponding to the zero-transition thickness. Description about this issue is added in Supplementary Information (page 2).

Comment 5:

The manuscript does not show or describe the temporal evolution of the magnetic field in much detail. For example, how do the Gauss coefficients vary with time and how does this compare to observations? What is the standard deviation of the magnetic spectra shown in

Figure 2e-f?

Reply:

According to reviewer's suggestion, time-series of the Gauss coefficients for the axial dipole, axial quadrupole, and axial dipole offset, D_{off} , were displayed in Fig. S7. They show very slow time variation. Error bars at one standard deviation level were added to the plots of magnetic power spectra (Figure 2e, 2f).

Comment 6:

The thickness of the mantle, which is needed to extrapolate the dynamo simulation results to the planet surface, is not specified in the text.

Reply:

Following Christensen (2006), Christensen and Wicht (2008), and Manglik et al. (2010) taking the Mercury's radius of 2440 km and the core radius of 1850 km, the thickness of the mantle is assumed to be 590 km. The assumed thickness of the mantle is shown in the caption of Figure 2 in the revised manuscript.

Comment 7:

No discussion on the impact of assuming (necessarily) high Ekman and magnetic Prandtl numbers and a low Lewis number was included.

Reply:

Some words for the necessity of exploration for these parameters were added in the revised version (lines 153-157).

- Conclusions:

Comment 8:

The manuscript argues that thermal convection due to internal heating is preferred in Mercury's core and, therefore, that a considerable amount of radioactive heat sources are required. Is this consistent with thermal evolution models and compositional constraints?

Reply:

Although thermal evolution history and chemical composition in Mercury's interior is still poorly constrained, our implications requiring radioactive heat source are consistent with calculations of thermal evolution and chemical composition such as Williams et al. (2007) and

Malavergne et al. (2010) (refs. 31 and 32 in the revised manuscript).

Comment 9:

The last paragraph is very speculative and necessitates additional simulations with different thicknesses of the stably stratified layer.

Reply:

We rewrote the last paragraph following the reviewer's comment.

- Suggested improvements:

Comment 10:

I recommend addressing the temporal evolution of the magnetic fields obtained in more detail. A discussion of dipole tilt angle in addition to F_{axis} would enable further comparison with observations and the broader literature. Inclusion of the local Rossby number (e.g., Christensen and Aubert, GJI, 2006) would also help characterize the flow.

Reply:

According to the reviewer's suggestion, some words regarding the dipole tilt angles were added in the revised manuscript (lines 91-93). As described in reply to comment 5, to show time variation, time-series of the Gauss coefficients for the axial dipole, axial quadrupole, and dipole offset, D_{off} , were added in Fig. S7. Also, the mean dipole tilt angle and the local Rossby number were included in Table S1.

Comment 11:

It would be helpful to further distinguish the differences between these models and those of Manglik et al. (2010). The radial heat and composition flux profiles are very different, and this should be highlighted more since the other input parameters are quite similar. For example, adding panels to Figure S1 for the high-sulphur and low-sulphur cases from Manglik et al. (2010), with similar non-dimensionalizations, would enable a better comparison between the models.

Reply:

According to the reviewer's suggestion, those profiles corresponding to Manglik et al. (2010) were added as Figure S4.

Comments 12:

Similarly, the authors could better address why magnetic field feedback on the flow is necessary to generate the helicity asymmetry. Sreenivasan et al. (2014), Soderlund et al. (2015), and Aurnou and King (2017) may be helpful in this regard.

*Sreenivasan, B., Sahoo, S., & Dhama, G. (2014). The role of buoyancy in polarity reversals of the geodynamo. *Geophysical Journal International*, 199(3), 1698-1708.*

*Soderlund, K. M., Sheyko, A., King, E. M., & Aurnou, J. M. (2015). The competition between Lorentz and Coriolis forces in planetary dynamos. *Progress in Earth and Planetary Science*, 2(1), 24.*

*Aurnou, J. M., & King, E. M. (2017). The cross-over to magnetostrophic convection in planetary dynamo systems. In *Proc. R. Soc. A* 473(2199), 20160731. The Royal Society.*

Reply:

According to the reviewer's suggestion, we added some sentences to address the necessity of the magnetic feedback (lines 138-153). Sreenivasan et al. (2014) was cited.

- References:

Comment 13:

Knibbe and van Westrenen (2018) would likely be of interest to the authors since it discusses the thermal and magnetic evolution of Mercury assuming an Fe-Si core and implements a conductive temperature profile in the core:

*Knibbe, J. S., & van Westrenen, W. (2018). The thermal evolution of Mercury's Fe-Si core. *Earth and Planetary Science Letters*, 482, 147-159.*

Reply:

According to the reviewer's suggestion. Some words for discussion were added citing Knibbe and van Westrenen (2018) in the revised manuscript (lines 189-191). Thanks.

Comment 14:

Landeau and Aubert (2011) find hemispheric dynamos with equatorially asymmetric convection and would be appropriate to cite:

*Landeau, M., & Aubert, J. (2011). Equatorially asymmetric convection inducing a hemispherical magnetic field in rotating spheres and implications for the past martian dynamo. *Physics of the Earth and planetary interiors*, 185(3-4), 61-73.*

Reply:

According to the reviewer's suggestion, Landeau and Aubert (2011) was cited and discussed

(lines 164-169).

Comment 15:

- Clarity and context:

The manuscript is not particularly well written. The models are not clearly described in the main text; it would be helpful to at least state how many models were run and what's varied between them.

Reply:

A few words were added to the revised version following the reviewer's comment (lines 74-76).

Other changes

Title is slightly changed according to the comment by reviewer#1.

Figure S1f was replaced, because the profiles for SL2 were not drawn in the original version.

Reviewers' comments:

Reviewer #1 (Remarks to the Author):

The author responded in a partially satisfactory way to the points raised by me. While the paper would be a useful addition to the scientific literature on Mercury's magnetic field in its present form, a few additional changes would improve it. I would leave this for the authors to consider.

Specifically, following the numbering of my previous comments used in the authors' reply:

1) To me the mutual compatibility of internal heating and a thermally stable layer in the upper part of Mercury's liquid core is still an open issue. True, a detailed thermodynamic analysis of this question is beyond the scope of this paper. Nonetheless, some qualitative discussion would be helpful to give a sense of how likely or plausible the whole concept of the BU-series of the models is.

2) Ok

3) The English still needs improvement in several places.

4) Adding the tilt angle to table S1 is useful. However, because it is claimed that some of the models match >>all<< essential properties of Mercury's magnetic field (strength, axisymmetry, dipole offset), also some measure of the observable field strength, such as the g_{10} coefficient or the rms surface strength, should be added to table S1.

Quoting values of Λ in the interior of the dynamo is not enough, because it cannot be compared to the observed field strength.

5) It may well be that a double-diffusive treatment is needed for the success of models such as BU1 or BU2, but it is not convincingly shown. It is not true that a combination of different locations of buoyancy sources (bottom / internal) cannot be treated in a codensity context, which is seemingly argued for in the reply to my comment. The question is whether different diffusivities must necessarily be associated with the two buoyancy sources. Of course, when both thermal and compositional buoyancy play a role, the diffusivities will be different. However, because double-diffusive convection is more complex, it would be very interesting to know if such treatment is indispensable to explain the observations (Occam's razor). This could be tested without too much effort, for example by running a case similar to BU1, but setting also the compositional Prandtl number to 0.1, i.e. the same value as the thermal Prandtl number. This is essential identical to a codensity approach.

6) Ok. With the smaller dipole offset inferred in the work by Thebault et al., it is also justified to concentrate on case BU1 rather than BU2.

7) Ok

8) Ok

9) Ok

Reviewer #2 (Remarks to the Author):

The authors have taken into account most of my remarks and gave reasonable answers to my questions and concerns.

I think the discussion is now stronger, and the results have gained significance.

I therefore recommend publication.

There is one last thing: the new text (in red) is sometimes difficult to read/follow because of the english.

I'm not a native english speaker, but I think you should try to improve the english of the new text.

Reviewer #3 (Remarks to the Author):

My primary concerns with the initial manuscript were (i) the lack of novelty for publication in Nature, (ii) insufficient explanations for the observed behaviors, and (iii) lack of discussion regarding the temporal variation of the simulated magnetic fields.

The revised manuscript does a good job addressing concern (iii), and concern (i) was largely alleviated. However, I continue to have significant concerns regarding (ii) as detailed in the blue text below. The manuscript is still not written well, and the new text is especially problematic.

Replies to Reviewer #3

First of all, the authors would like to thank the reviewer for her/his valuable comments, which undoubtedly help us improve the manuscript.

In the following, reviewer's comments are written in bold italic.

- Originality and significance:

Comment 1:

The role of an outer stable layer, double-diffusive convection, and bottom-up versus top-down versus snow layer convective modes have all been studied previously in the context of planetary dynamos, including for Mercury in particular (e.g., Christensen 2006; Christensen and Wicht 2008; Stanley and Mohammadi 2008; Manglik et al. 2010; Vilim et al. 2010; Cao et al. 2014; Tian et al. 2015). While this manuscript considers some new combinations of these elements to reproduce key elements of Mercury's magnetic field (dipole offset, strong axisymmetry, weak field strength) simultaneously for the first time, I am concerned that the study is not sufficiently novel for publication here.

Author Reply:

We hope our point-by-point responses given below will alleviate the reviewer's concern about our results.

Comment 2:

The addition of double diffusion to these models causes the magnetic field strength to increase due to flows that are generated in the thermally stratified layer (Manglik et al. 2010), although these authors note that a double diffusive dynamo that matches the observed field strength could likely be found if the input parameters were tuned in their low compositional buoyancy flux case. The manuscript would benefit from a more explicit discussion of how their

simulations extend these results (see also later comment). Similarly, the manuscript does not explain if (or why) double-diffusive convection is a necessary ingredient for generating a Mercury-like field or if a co-density approach would likely be adequate.

Author Reply:

Double-diffusive convection modelling enables us to treat thermal and compositional buoyancy separately applying different diffusivity coefficients. Hence, we can distinctly put buoyancy source at the ICB for compositional convection, and within the core for thermal convection. It is a great advantage of this treatment, whereas the co-density formulation cannot tell the difference in principle. As the reviewer pointed out below, contribution from a volumetrically distributed source seems to be necessary for a Mercury-like dynamo. Hence, we think separation between bottom-driven and internally-driven convection is a key element for the present results. Moreover, considering double-diffusive convection could provide various implications for thermal evolution, and chemical composition of Mercury as well as core dynamics. We added discussion with previous results, and some words for the necessity of double-diffusive treatment in the revised version (lines 158-174).

Reviewer Reply:

The new text added on lines 134-137 and Figures S4 and S6 are helpful towards distinguishing against the Manglik et al. (2010) results. However, I'm still left with the questions of why an axisymmetric toroidal field develops around the stratification boundary in BU1, why it does not develop in the other models, and why it is important for generation of the dipole field component. The authors state that this latter point will be examined in future works, but I consider it to be important for this study.

The new text on lines 158-174 is poorly written. In addition, these arguments take the defensive approach that other models don't work as well as those presented rather than an assertive approach explaining why double diffusive convection leads to a specific flow field that is responsible for generating a weak, strongly axisymmetric, equatorially asymmetric magnetic field with weak secular variation for the following reasons. The new Supplementary Figures help towards this end, but are not discussed in any detail or at all (Figures S5 and S9).

Furthermore, Christensen (Icarus, 2015) finds weak, axisymmetric, hemispheric magnetic fields using a co-density approach when a thick, strongly stratified outer layer is present in their dynamo models for Ganymede. Their Figure 12 also demonstrates that the field morphology varies with stable layer thickness and input parameters. This may help explain why case BU3, which has

lower Rayleigh numbers, is dipole-dominated with a small offset. It also suggests that different results may be obtained, and the conclusions modified, if the stable layer thickness were varied.

Comment 3:

Cao et al. (2014) considered volumetrically distributed versus bottom-driven convection and concluded that the former is a necessary ingredient for axial-dominant, equatorially asymmetric dynamos. Local excess equatorial CMB heat flow causes these solutions to be steady in time. They further argue that this magnetic field morphology is the result of equatorially asymmetric helicity that results from the mutual excitation of two different modes of columnar convection, similar to this manuscript.

The role of magnetic field feedback on the maintenance of equatorially asymmetric helicity is novel and very interesting. However, this result is not discussed sufficiently (see also later comment). Given the weak magnetic field strengths of the models, why does this feedback occur? Is this phenomena unique to Mercury? If so, why?

Author Reply:

We agree with the reviewer's argument about the results by Cao et al. (2014), which is seemingly similar to our results. As the reviewer mentioned, an important point in the present results is that the Lorentz force plays a critical role in yielding and maintaining an equatorially asymmetric helicity distribution. Mechanisms responsible for generating and maintaining helicity asymmetry here is not explained by a linear superposition of the symmetric and antisymmetric velocity fields as in Cao et al. (2014), but by the non-linear interaction between two different modes prompted by feedback effects of the Lorentz force, which is evidently demonstrated by our kinematic run. We consider the magnetic field within the convective region is strong enough for this feedback to work, in particular near the stable and unstable stratification boundary, whereas the weak magnetic field at the surface is due to attenuation associated with skin effects of the stably stratified layer. In order to show them clearly, we added plots of the rms-magnetic field profiles with respect to the radius, and time-series of the kinetic energy in the antisymmetric component in the kinematic run as Figs. S6 and S8.

Based on the present results, it is found that distinct distribution of thermo-compositional buoyancy source as well as a thick stably stratified layer is important to reproduce Mercury-like morphology and strength of the magnetic field. Although the magnetic feedback should generally occur, how effectively it works may depend on some conditions such as buoyancy source distribution, thickness of the stably stratified layer, inner core size and so on, all of which reflect thermo-chemical evolution of the planets. In this sense, the observable feedback may be specific to Mercury's core. It is very interesting and important to examine difference with a body similar

to Mercury in size such as Ganymede, which remains as a future work.

Reviewer Response:

The new text the better distinguishes against the results of Cao et al. (2014), and the new Figure S8 clearly demonstrates how the anti-symmetric mode decays in the kinematic run. The discussion above in regards to Figure S6 should also be added to the manuscript.

- Data, methodology, statistics, and uncertainties:

Comment 4:

The thermal and compositional profiles used in these models appear to be a critical component for generating Mercury-like magnetic fields, and are different from those employed in previous models for Mercury's dynamo. These profiles are described in detail, but not justified in a physical way. In particular, the motivation and realism of tanh functions are not explained sufficiently, especially in the context of radial interior structure/temperature/composition estimates.

Author Reply:

Since the radial profiles of temperature and composition in Mercury's core are not well constrained at present as far as we know, we choose the heat flux profile to ensure stable stratification in the upper part of the core. In this sense, using the form of hyperbolic tangent is for mathematical convenience, which allows a continuous change in heat flux between convecting and stable regions as in, for example, Takehiro and Lister (2001). This formulation allows us to put the stably stratified region without seriously affecting the radial profile outside the stable layer (effects are limited within transition thickness given as a parameter), and thus to easily see effects of the stably stratified layer with different thickness on convection and resultantly the magnetic field. In some previous studies, an abrupt, discontinuous change in heat or co-density flux is imposed to implement the stably stratified layer (e.g. Stanley and Mohammadi, PEPI, 2008; Vilm et al., JGR, 2010). The present way is an extension of these corresponding to the zero-transition thickness. Description about this issue is added in Supplementary Information (page 2).

Comment 5:

The manuscript does not show or describe the temporal evolution of the magnetic field in much detail. For example, how do the Gauss coefficients vary with time and how does this compare to observations? What is the standard deviation of the magnetic spectra shown in Figure 2e-f?

Author Reply:

According to reviewer's suggestion, time-series of the Gauss coefficients for the axial dipole, axial quadrupole, and axial dipole offset, D_{off} , were displayed in Fig. S7. They show very slow time variation. Error bars at one standard deviation level were added to the plots of magnetic power spectra (Figure 2e, 2f).

Comment 6:

The thickness of the mantle, which is needed to extrapolate the dynamo simulation results to the planet surface, is not specified in the text.

Author Reply:

Following Christensen (2006), Christensen and Wicht (2008), and Manglik et al. (2010) taking the Mercury's radius of 2440 km and the core radius of 1850 km, the thickness of the mantle is assumed to be 590 km. The assumed thickness of the mantle is shown in the caption of Figure 2 in the revised manuscript.

Comment 7:

No discussion on the impact of assuming (necessarily) high Ekman and magnetic Prandtl numbers and a low Lewis number was included.

Author Reply:

Some words for the necessity of exploration for these parameters were added in the revised version (lines 153-157).

Reviewer Reply:

While I appreciate that realistic values cannot be simulated, it is not convincing to simply conjecture that the effects could work to some extent. This argument should be improved.

- Conclusions:

Comment 8:

The manuscript argues that thermal convection due to internal heating is preferred in Mercury's core and, therefore, that a considerable amount of radioactive heat sources are required. Is this consistent with thermal evolution models and compositional constraints?

Author Reply:

Although thermal evolution history and chemical composition in Mercury's interior is still poorly constrained, our implications requiring radioactive heat source are consistent with calculations of

thermal evolution and chemical composition such as Williams et al. (2007) and Malavergne et al. (2010) (refs. 31 and 32 in the revised manuscript).

Comment 9:

The last paragraph is very speculative and necessitates additional simulations with different thicknesses of the stably stratified layer.

Author Reply:

We rewrote the last paragraph following the reviewer's comment.

Reviewer Reply:

The last paragraph is still too speculative without additional simulations with different thicknesses of the stably stratified layer.

- Suggested improvements:

Comment 10:

I recommend addressing the temporal evolution of the magnetic fields obtained in more detail. A discussion of dipole tilt angle in addition to F_{axs} would enable further comparison with observations and the broader literature. Inclusion of the local Rossby number (e.g., Christensen and Aubert, GJI, 2006) would also help characterize the flow.

Author Reply:

According to the reviewer's suggestion, some words regarding the dipole tilt angles were added in the revised manuscript (lines 91-93). As described in reply to comment 5, to show time variation, time-series of the Gauss coefficients for the axial dipole, axial quadrupole, and dipole offset, D_{off} , were added in Fig. S7. Also, the mean dipole tilt angle and the local Rossby number were included in Table S1.

Comment 11:

It would be helpful to further distinguish the differences between these models and those of Manglik et al. (2010). The radial heat and composition flux profiles are very different, and this should be highlighted more since the other input parameters are quite similar. For example, adding panels to Figure S1 for the high-sulphur and low-sulphur cases from Manglik et al. (2010), with similar non-dimensionalizations, would enable a better comparison between the models.

Author Reply:

According to the reviewer's suggestion, those profiles corresponding to Manglik et al. (2010) were added as Figure S4.

Comments 12:

Similarly, the authors could better address why magnetic field feedback on the flow is necessary to generate the helicity asymmetry. Sreenivasan et al. (2014), Soderlund et al. (2015), and Aurnou and King (2017) may be helpful in this regard.

Sreenivasan, B., Sahoo, S., & Dhama, G. (2014). The role of buoyancy in polarity reversals of the geodynamo. Geophysical Journal International, 199(3), 1698-1708.

Soderlund, K. M., Sheyko, A., King, E. M., & Aurnou, J. M. (2015). The competition between Lorentz and Coriolis forces in planetary dynamos. Progress in Earth and Planetary Science, 2(1), 24.

Aurnou, J. M., & King, E. M. (2017). The cross-over to magnetostrophic convection in planetary dynamo systems. In Proc. R. Soc. A 473(2199), 20160731. The Royal Society.

Author Reply:

According to the reviewer's suggestion, we added some sentences to address the necessity of the magnetic feedback (lines 138-153). Sreenivasan et al. (2014) was cited.

- References:

Comment 13:

Knibbe and van Westrenen (2018) would likely be of interest to the authors since it discusses the thermal and magnetic evolution of Mercury assuming an Fe-Si core and implements a conductive temperature profile in the core:

Knibbe, J. S., & van Westrenen, W. (2018). The thermal evolution of Mercury's Fe-Si core. Earth and Planetary Science Letters, 482, 147-159.

Author Reply:

According to the reviewer's suggestion. Some words for discussion were added citing Knibbe and van Westrenen (2018) in the revised manuscript (lines 189-191). Thanks.

Comment 14:

Landeau and Aubert (2011) find hemispheric dynamos with equatorially asymmetric convection and would be appropriate to cite:

Landeau, M., & Aubert, J. (2011). Equatorially asymmetric convection inducing a

hemispherical magnetic field in rotating spheres and implications for the past martian dynamo. Physics of the Earth and planetary interiors, 185(3-4), 61-73.

Author Reply:

According to the reviewer's suggestion, Landeau and Aubert (2011) was cited and discussed (lines 164-169).

Comment 15:

- Clarity and context:

The manuscript is not particularly well written. The models are not clearly described in the main text; it would be helpful to at least state how many models were run and what's varied between them.

Author Reply:

A few words were added to the revised version following the reviewer's comment (lines 74-76).

Reviewer Reply:

The manuscript still suffers from poor English throughout.

Other changes

Title is slightly changed according to the comment by reviewer#1.

Figure S1f was replaced, because the profiles for SL2 were not drawn in the original version.

In the last sentence of the revised Figure 4 caption, why does the case being supercritical matter for the units?

Replies to Reviewer #1

First of all, the authors would like to thank the reviewer for her/his valuable and constructive comments, which undoubtedly help us improve the manuscript.

In the following, reviewer's comments are written in bold italic.

1) To me the mutual compatibility of internal heating and a thermally stable layer in the upper part of Mercury's liquid core is still an open issue. True, a detailed thermodynamic analysis of this question is beyond the scope of this paper. Nonetheless, some qualitative discussion would be helpful to give a sense of how likely or plausible the whole concept of the BU-series of the models is.

Reply:

According to the reviewer's comment, a discussion on the mutual compatibility of internal heating and a thermally stably stratified layer were added (lines 220 – 235). It is based on reviewer's suggestion in the first reviewing process.

3) The English still needs improvement in several places.

Reply:

According to the reviewer's comment, we carefully checked the manuscript with the aid of English editing service. We hope the English is improved in a satisfactory way.

4) Adding the tilt angle to table S1 is useful. However, because it is claimed that some of the models match >>all<< essential properties of Mercury's magnetic field (strength, axisymmetry, dipole offset), also some measure of the observable field strength, such as the g_{10} coefficient or the rms surface strength, should be added to table S1.

Quoting values of Λ in the interior of the dynamo is not enough, because it cannot be compared to the observed field strength.

Reply:

We added the g_{10} coefficient in table S1 according to the reviewer's comment.

5) It may well be that a double-diffusive treatment is needed for the success of models such as BU1 or BU2, but it is not convincingly shown. It is not true that a combination of different

locations of buoyancy sources (bottom / internal) cannot be treated in a codensity context, which is seemingly argued for in the reply to my comment. The question is whether different diffusivities must necessarily be associated with the two buoyancy sources. Of course, when both thermal and compositional buoyancy play a role, the diffusivities will be different. However, because double-diffusive convection is more complex, it would be very interesting to know if such treatment is indispensable to explain the observations (Occam's razor). This could be tested without too much effort, for example by running a case similar to BU1, but setting also the compositional Prandtl number to 0.1, i.e. the same value as the thermal Prandtl number. This is essential identical to a codensity approach.

Reply:

Thank you for very helpful and constructive comment. Following the reviewer's suggestion, we performed an additional run with the compositional Prandtl number set to 0.1. In the run, any significant dipole offset does not appear, and the resultant velocity/magnetic field structures are strongly symmetric/anti-symmetric with respect to equator. It is also confirmed that the self-regulation effects work to some extent, but not enough to form dipole offset, even in this co-density case, since fraction of the antisymmetric flow components are non-zero. Thus, it is demonstrated, we think, that double-diffusive approach is a necessary ingredient for the present results, although physical details remain to be resolved with a systematic parameter survey. Sentences regarding the co-density run were added in the revised manuscript (lines 183-191).

Replies to Reviewer #2

First of all, the authors would like to thank the reviewer (Dr. N. Schaeffer) for his valuable comments, which undoubtedly help us improve the manuscript.

In the following, reviewer's comments are written in bold italic.

There is one last thing: the new text (in red) is sometimes difficult to read/follow because of the english.

I'm not a native english speaker, but I think you should try to improve the english of the new text.

Reply:

According to the reviewer's comment, we carefully check the manuscript to improve the English with the aid of English editing service. We hope the English is improved in a satisfactory way.

Replies to Reviewer #3

First of all, the authors would like to thank the reviewer for her/his valuable comments, which undoubtedly help us improve the manuscript.

In the following, reviewer's comments are written in bold italic.

We have divided the Comment 2 into two parts as follows.

Comment 2.1:

The new text added on lines 134-137 and Figures S4 and S6 are helpful towards distinguishing against the Manglik et al. (2010) results. However, I'm still left with the questions of why an axisymmetric toroidal field develops around the stratification boundary in BU1, why it does not develop in the other models, and why it is important for generation of the dipole field component. The authors state that this latter point will be examined in future works, but I consider it to be important for this study.

Reply:

We have added Figures 5 and S10 to clarify the magnetic and velocity fields' structures and generation processes of the axial dipole, the axial quadrupole and the axisymmetric toroidal field, in each model. A discussion on this point is given in the revised manuscript (lines 163 – 181).

Comment 2.2:

The new text on lines 158-174 is poorly written. In addition, these arguments take the defensive approach that other models don't work as well as those presented rather than an assertive approach explaining why double diffusive convection leads to a specific flow field that is responsible for generating a weak, strongly axisymmetric, equatorially asymmetric magnetic field with weak secular variation for the following reasons. The new Supplementary Figures help towards this end, but are not discussed in any detail or at all (Figures S5 and S9).

Furthermore, Christensen (Icarus, 2015) finds weak, axisymmetric, hemispheric magnetic fields using a co-density approach when a thick, strongly stratified outer layer is present in their dynamo models for Ganymede. Their Figure 12 also demonstrates that the field morphology varies with stable layer thickness and input parameters. This may help explain why case BU3, which has lower Rayleigh numbers, is dipole-dominated with a small offset. It also suggests that different results may be obtained, and the conclusions modified, if the

stable layer thickness were varied.

Reply:

We rewrote this part carefully considering the reviewer's comment. Please note that most of this paragraph was devoted for discussion to point out differences between the present results and previous studies according to the other reviewer's comment. To test if double diffusive convection is really necessary for generating a Mercury-like field, we carried out an additional run corresponding to the co-density case. Consequently, the co-density run fails to generate the Mercury-like morphology (we added a Figure S11). This demonstrates the necessity of double diffusive convection for the axial-dipole offset. At present, however, it is difficult to find a definite answer to the question how double diffusive convection plays a role in generating the equatorially-asymmetric magnetic field, because double diffusive convection includes additional complex processes, which have been poorly studied and understood in a problem of spherical shell dynamo. This is exactly a future work. Some words regarding Figures S5 and S9 as well as new Figures 5, S10, and S11 were added in the revised version.

We are interested in the results of Christensen (2015). However, it is not clear what mechanism is responsible for formation of hemispherical morphology in his results, although it is suggested that vigorous convection and a thick stably stratified layer may be important. The comment is also related to Comment 9 and partly Comment 7 below. Concerning the stable layer thickness, the reviewer's comment is almost identical with what we have meant in the last paragraph of the previous version of the manuscript. If runs with different stable layer thickness would yield different results especially in field morphology, then they may somehow help to constrain the stable layer thickness using the field morphology consistent with the observations. Obviously, we need many more simulation results in a wide range of parameter space. We mentioned this in the revised version (lines 203-210). Also, taking into account the fact that this is far beyond the scope of the present paper and the reviewer's Comment 9 below, we rewrote the last paragraph in a more conservative way than in the previous manuscript.

Comment 3:

The new text the better distinguishes against the results of Cao et al. (2014), and the new Figure S8 clearly demonstrates how the anti-symmetric mode decays in the kinematic run. The discussion above in regards to Figure S6 should also be added to the manuscript.

Reply:

According to the reviewer's suggestion, discussion given in our reply were added to the revised manuscript (lines 142-145).

Comment 7:

While I appreciate that realistic values cannot be simulated, it is not convincing to simply conjecture that the effects could work to some extent. This argument should be improved.

Reply:

As the reviewer pointed out, we need a parameter survey to say something convincingly. We rewrote this part in the revised version (lines 205-210).

Comment 9:

The last paragraph is still too speculative without additional simulations with different thicknesses of the stably stratified layer.

Reply:

As in our reply to Comment 2.2 above, we rewrote the paragraph more conservatively according to the reviewer's comment.

Comment 15:

The manuscript still suffers from poor English throughout.

Reply:

According to reviewer's comment, we carefully check the manuscript to improve the English with the aid of English editing service. We hope the English is improved in a satisfactory way.

Others:

In the last sentence of the revised Figure 4 caption, why does the case being supercritical matter for the units?

Reply:

In this supercritical case, the magnetic field grows in an exponential fashion with time due to lack of the feedback effects of the Lorentz force on the velocity field. The magnetic field is, thus, normalized, since actual numerical values in the magnetic field have no physical meaning.

REVIEWERS' COMMENTS:

Reviewer #1 (Remarks to the Author):

The authors' reaction to all my points is completely satisfactory.

Reviewer #3 (Remarks to the Author):

My primary concerns with the revised manuscript were regarding insufficient explanations for the observed behaviors. The new revisions generally did a good job addressing my comments or noting that they would be addressed in future work.

Minor comments:

- Line 118: "perfectly" is too strong
- Lines 223-238: These arguments would be strengthened by adding some references.
- Figure 5 caption: Specify what the magnetic field plots are normalized by.
- Supplement, page 2: Add a reference for why the inner core can be treated as an insulator (e.g., Wicht, PEPI, 2002).
- Figure S10: Specify whether the cross sections are time-averaged or snapshots in time, and define the black circles as done in other figures (e.g., Fig S11).

Reply to Reviewer #1

In the following, reviewer's comments are written in bold italic.

The authors' reaction to all my points is completely satisfactory.

The authors would like to thank the reviewer for her/his valuable and constructive comments, which undoubtedly helped us improve the manuscript.

Reply to Reviewer #3

In the following, reviewer's comments are written in bold italic.

My primary concerns with the revised manuscript were regarding insufficient explanations for the observed behaviors. The new revisions generally did a good job addressing my comments or noting that they would be addressed in future work.

The authors would like to thank the reviewer for her/his valuable and constructive comments, which undoubtedly helped us improve the manuscript.

Minor comments:

- Line 118: "perfectly" is too strong

We changed the wording to "nearly-perfectly" according to reviewer's suggestion (Line 114).

- Lines 223-238: These arguments would be strengthened by adding some references.

We cited Dumberry and Rivoldini (2015) and Knibbe and van Westrenen (2018) (refs. 14 and 33) according to the reviewer's suggestion (Line 231).

- Figure 5 caption: Specify what the magnetic field plots are normalized by.

We clearly mentioned that the magnetic field plots are normalized by the maximum values, following the reviewer's comment.

- Supplement, page 2: Add a reference for why the inner core can be treated as an insulator (e.g., Wicht, PEPI, 2002).

According to the reviewer's suggestion, Wicht (2002) was cited in Methods (Line 289). Most part of Supplementary Information was moved to Methods following the instruction by the Editor.

- Figure S10: Specify whether the cross sections are time-averaged or snapshots in time, and define the black circles as done in other figures (e.g., Fig S11).

Time-averaged plots are shown in the cross-section. We rewrote the caption according to the reviewer's suggestion.